# Simultaneous inhibition of DNA-PK and PolΘ improves integration efficiency and precision of genome editing

Sandra Wimberger [1,2] ✉, Nina Akrap[1], Mike Firth[3], Johan Brengdahl[4], Susanna Engberg[5], Marie K. Schwinn[6], Michael R. Slater [6], Anders Lundin [7], Pei-Pei Hsieh[1], Songyuan Li[1], Silvia Cerboni[8], Jonathan Sumner[9], Burcu Bestas [1], Bastian Schiffthaler[10], Björn Magnusson[7], Silvio Di Castro[11], Preeti Iyer[12], Mohammad Bohlooly-Y[7], Thomas Machleidt[6], Steve Rees[13], Ola Engkvist [12], Tyrell Norris[5], Elaine B. Cadogan [14], Josep V. Forment[14], Saša Šviković [1], Pinar Akcakaya[1], Amir Taheri-Ghahfarokhi[1] & Marcello Maresca [1] ✉

Genome editing, specifically CRISPR/Cas9 technology, has revolutionized biomedical research and offers potential cures for genetic diseases. Despite rapid progress, low efficiency of targeted DNA integration and generation of unintended mutations represent major limitations for genome editing applications caused by the interplay with DNA double-strand break repair pathways. To address this, we conduct a large-scale compound library screen to identify targets for enhancing targeted genome insertions. Our study reveals DNA-dependent protein kinase (DNA-PK) as the most effective target to improve CRISPR/Cas9-mediated insertions, confirming previous findings. We extensively characterize AZD7648, a selective DNA-PK inhibitor, and find it to significantly enhance precise gene editing. We further improve integration efficiency and precision by inhibiting DNA polymerase theta (PolΘ). The combined treatment, named 2iHDR, boosts templated insertions to 80% efficiency with minimal unintended insertions and deletions. Notably, 2iHDR also reduces off-target effects of Cas9, greatly enhancing the fidelity and performance of CRISPR/Cas9 gene editing.

Clustered Regularly Interspaced Short Palindromic Repeats - CRISPR-associated protein 9 (CRISPR-Cas9) genome editing is a widely applied genome-level manipulation strategy that has become an indispensable tool for biomedical research. Recent pre-clinical and clinical studies have demonstrated the potential for CRISPR-Cas9 to treat genetic disorders[1–3]. Currently, the most widely used genome editing strategy is based on a single guide RNA (sgRNA) directed DNA double-strand break (DSB) mediated by *Streptococcus pyogenes* Cas9 endonuclease (*Sp*Cas9). The DSB caused by Cas9 triggers the activation of several cellular DSB repair pathways[4], with potentially distinct gene editing outcomes. Thus, precise gene editing is achieved by hijacking the homology-directed repair (HDR) pathway, which employs a homologous DNA template for high-fidelity repair. Typically, cells use the sister chromatid DNA as a repair template for HDR[5]. However, in gene editing applications, an exogenous donor DNA is used to outcompete this process and introduce desired edits at the integration site. Unfortunately, HDR-dependent genome editing in mammalian cells is limited due to the slow HDR repair kinetics and its restriction to the S/G2 phase of the cell cycle[6,7]. Consequently, most modifications in an edited cell population represent insertions and/or deletions (InDels) due to preferred usage of the competing DNA end-joining repair pathways, including non-homologous end joining (NHEJ) and alternative end joining (alt-EJ)[8].

NHEJ is the fastest and predominant DSB repair pathway that achieves the ligation of DNA ends without the need for a homologous template. A critical component of the NHEJ pathway is the DNA-dependent protein-kinase catalytic subunit (DNA-PKcs) that is recruited by the Ku heterodimer to DNA DSBs to form the DNA-PK complex, which undergoes auto-phosphorylation and activates additional NHEJ factors[9]. Subsequently, DNA ends are re-joined by DNA ligase 4 (LIG4) following minimal or no end processing, which generally results in seamless repair (restoring the target site) or generation of small InDels (<10 bp)[10,11]. In contrast, alt-EJ is inherently prone to errors and can potentially create larger deletions[12]. Alt-EJ typically uses short microhomology (MH) sequences flanking the break site and requires 3′-single-stranded DNA substrates, which are generated by nucleolytic processing of DNA, a process termed end resection. Though the initial steps of end resection are shared with HDR, limited end resection is sufficient to promote alt-EJ[13]. DNA polymerase theta (PolΘ) is the primary but not an exclusive mediator of alt-EJ in most eukaryotic cells. Therefore, alt-EJ through PolΘ is referred to as PolΘ-mediated end joining (TMEJ). The helicase domain of PolΘ most likely promotes the annealing of resected 3′ overhangs utilising microhomologies[14,15], while the polymerase domain extends annealed sequences. Resolution involves flap removal through endonucleases, such as Flap endonuclease 1 (FEN1)[16], gap filling, and, finally, joining of ends by DNA ligase 1 (LIG1) or DNA ligase 3 (LIG3)[14,17].

Although the imprecision of DSB repair pathways produces highly heterogenous repair outcomes, the "InDel profile" of NHEJ and alt-EJ repair is non-random, reproducible and to some extent predictable for a given target sequence[17–20]. Therefore, precise and predictable end joining repair has been successfully exploited for targeted integration and template-free correction of InDels[19,21–24]. However, the flexibility of HDR, in enabling scarless installation of various genomic modifications, such as point mutations, deletions and kilobase insertions, makes it an essential modality for research and clinical developments. Therefore, developing strategies to bias repair towards HDR is crucial for precise genome editing applications.

Accordingly, several studies have explored using small molecules to direct the repair pathway choice towards HDR[8]. For example, DNA-PK inhibitors have been shown to decrease NHEJ and therefore enhance HDR[25–27]. Recently developed potent and selective DNA-PK inhibitors facilitate higher HDR efficiencies and exhibit lower cytotoxicity compared to earlier compounds[28,29]. This observation indicates that using highly potent and selective DNA-PK inhibitors will expedite further improvements in genome editing applications. Although DNA-PK inhibitors result in increased HDR levels and reduction of NHEJ-associated InDels, MH-dependent deletions are still present and occasionally elevated upon DNA-PK inhibition[29–31]. Various reports have shown that knockout or knock-down of PolΘ partially reduces MH-associated deletions and minimises Cas9-related unwanted on-target effects, such as translocations and large deletions[30,32–36]. Therefore, these studies suggest that inhibition of PolΘ might contribute to improved gene targeting efficiencies and mitigate undesired on- and off-target effects, especially when combined with inhibition of DNA-PK activity. However, small molecule inhibitors of PolΘ have only recently started to emerge[37–39]. While we submitted this work for publication, a study was published demonstrating that the PolΘ inhibitor ART558[37,40] restrains the formation of large deletion and promotes HDR at CRISPR/Cas9-induced DSBs[41]. However, further investigations of other potential genomic alterations are still lacking. Identifying additional inhibitors with higher specificity and potency might further improve precise genome editing.

Here, we employed a fluorescence-based reporter assay to screen a library of 20,548 small molecules for compounds that increase precision of genome editing. Furthermore, we developed a marker-free analysis pipeline termed knock-in sequencing (KI-Seq) that enables studying and visualising DNA repair outcomes and knock-in (KI) strategies at endogenous loci. Using KI-Seq, we evaluated the ability of several compounds to alter DNA repair pathway choice. DNA-PK inhibitors are the most effective molecules to modulate editing precision in our compound library. We determine AZD7648 as the most potent and selective DNA-PK inhibitor, which reliably improves KI efficiencies in transformed and non-transformed cells. Finally, we show that using potent PolΘ inhibitors, targeting the polymerase or helicase domain of PolΘ, combined with AZD7648 further improve KI efficiencies. We demonstrate that these combinations, referred here as 2iHDR, enhance homology-based templated insertions, while reducing NHEJ/TMEJ-related mutagenic events and off-target editing.

## Results

### High-throughput screening platform for the discovery of compounds that increase HDR and decrease end joining repair

To identify potent small molecules that enhance HDR and inhibit end joining (EJ) repair we performed a high-throughput screening of a set of structurally diverse compounds. We utilised the previously published traffic light reporter (TLR)[42] for rapid measurement of DNA repair outcomes upon CRISPR-Cas9 DSB generation. In this system, the mutated enhanced green fluorescent protein (eGFP) coding sequence contains an in-frame stop codon and is linked to an out-of-frame (+2 bp) *Discosoma sp.* red fluorescent protein (DsRed) coding sequence. Co-expression of Cas9 nuclease in the presence of a sgRNA targeting the mutated eGFP sequence and a repair template will lead either to the restoration of eGFP expression or activation of DsRed expression through HDR or EJ-mediated repair, respectively. To facilitate the usage of the TLR system for high-throughput screening purposes, we generated a clonal HEK293-TLR line by stably integrating a single copy of the TLR construct into the *AAVS1* locus (Fig. 1a).

The screening library consisted of 20,548 small molecules and represents a diverse set of commercially available as well as internally generated tool compounds. Most of them (91%) have annotated targets and previously reported $pXC_{50}$ values (negative logarithm activity value of in vitro measured $IC_{50}$s, $EC_{50}$s, Ki, Kd or percent inhibition endpoints). Our library covered 1817 biological targets with reported $XC_{50}$ values below 100 nM. The main target classes were G–protein-coupled receptors, transporters and ion channels, metabolite inter-conversion enzymes, and protein kinases. The high-throughput screening workflow consisted of two steps: (1) a primary screen where all compounds were tested at 2 μM concentration; and (2) a hit confirmation screen where 380 selected compounds from the primary screen were tested to confirm their activity in a dose-dependent manner (Fig. 1b). The primary screen yielded 225 hits that increased EJ-mediated DNA repair, 16 that increased both EJ repair and HDR, 91 compounds that enhanced only HDR, and 73 that improved HDR while decreasing EJ repair (Fig. 1b, c). Next, we advanced these compounds into the hit confirmation screen. We identified 92 compounds that increased EJ-mediated DNA repair, three compounds that increased both EJ repair and HDR, 13 compounds that enhanced only HDR, and 43 compounds that improved HDR while decreasing EJ repair (Fig. 1b and Supplementary Fig. S1a). Next, we investigated the annotated targets of compounds that lead to improved HDR and decreased EJ repair and identified DNA-PK as the primary target for 13 of the compounds with measured pXC50 > 8.5 (Fig. 1b and Supplementary Fig. S1a). Taken together, our high-throughput screening of a diverse compound library identified DNA-PK inhibitors as potent regulators of DSB repair pathway selection that bias the pathway choice towards HDR, while disfavouring EJ repair.

### Knock-in sequencing identifies DNA repair outcomes and informs about suitable knock-in strategies

To further profile hits identified from the TLR screen through a comprehensive examination of DNA repair events at Cas9-induced DSBs, we developed KI-Seq, a next-generation sequencing (NGS)-based analysis pipeline of amplicon-sequencing (Amp-Seq) data that assigns

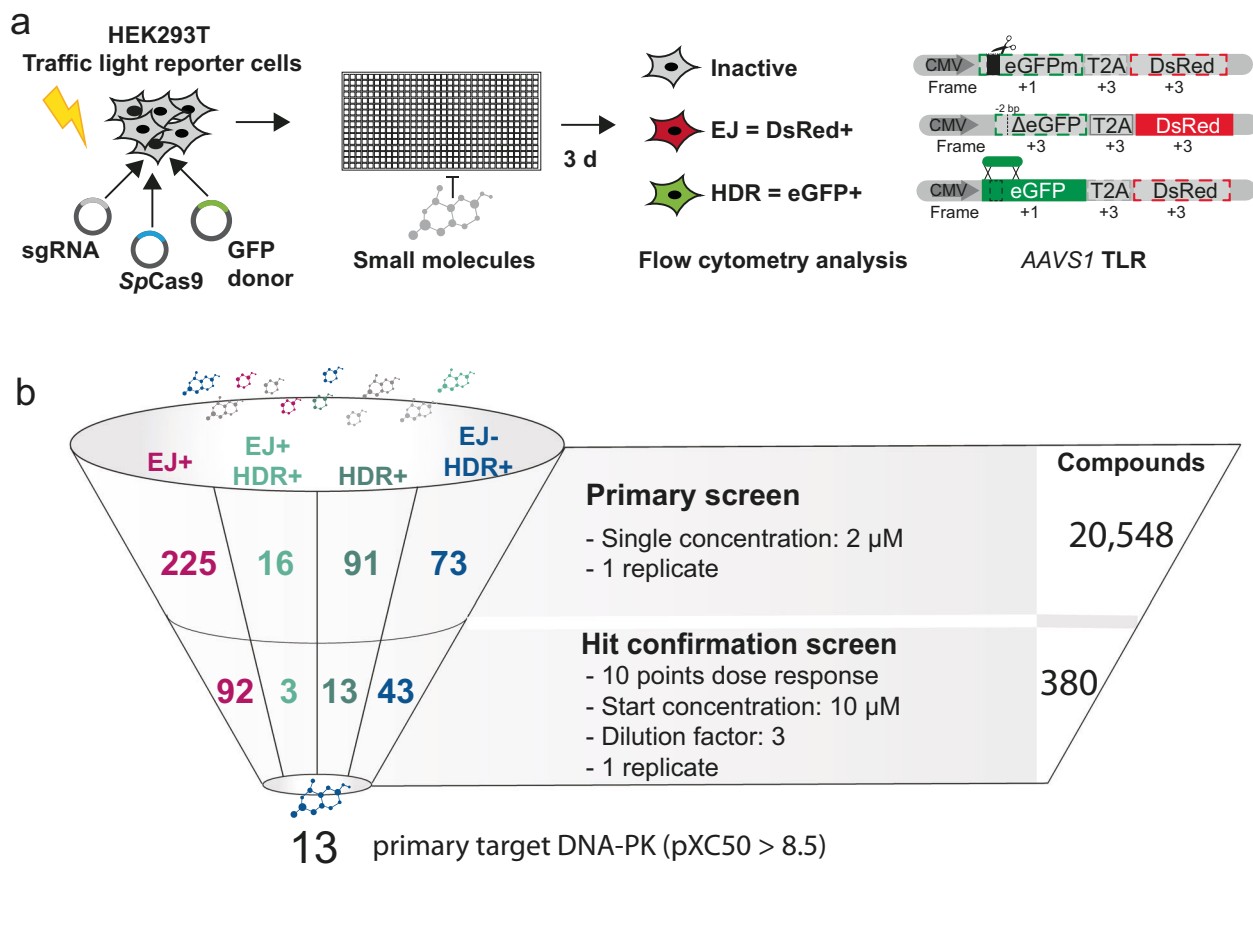

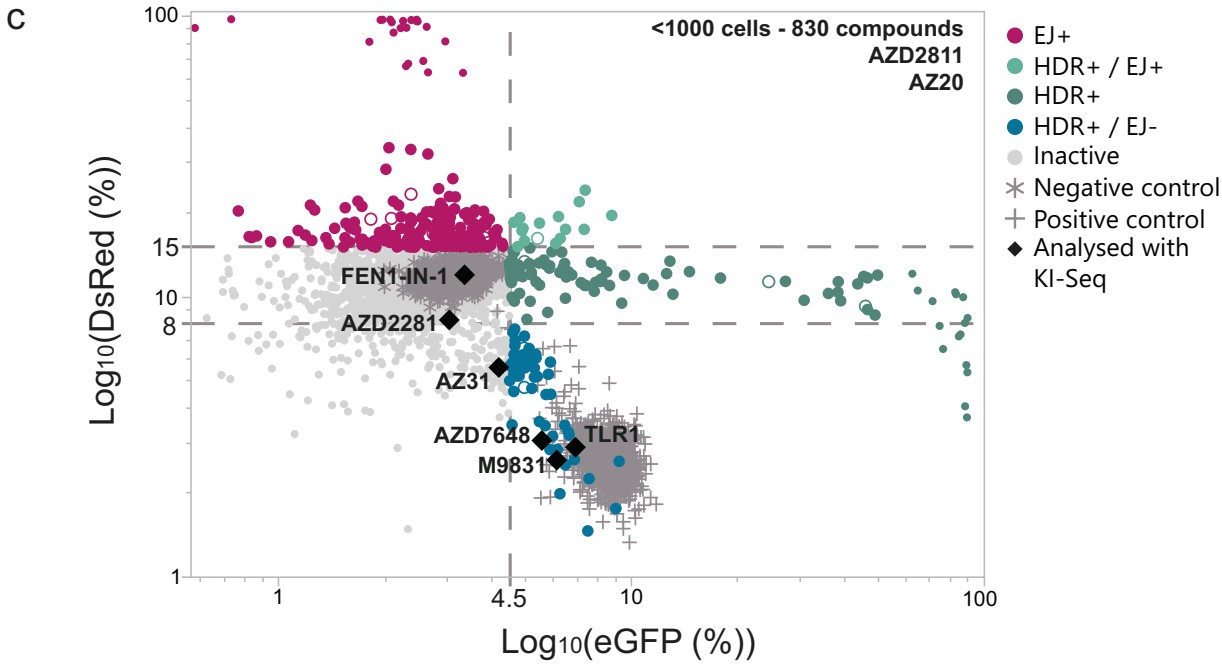

underlying DNA repair pathways to InDel profiles. KI-Seq uses CRISPResso2[43] and rational indel meta-analysis (RIMA) v2, an updated version of the previously described RIMA quantification method[34] with modifications that enable the analysis of Cas9-induced insertions resulting from HDR/NHEJ, as well as InDels from NHEJ/alt-EJ (Fig. 2a).

We selected three sgRNAs for our experiments based on RIMA analysis of a previous study[44]. We named them according to their main

InDel outcome, whereby gDel targeting *STAT1* induces a 1 bp deletion as the primary editing event, while gIns and gMej targeting *CD34* induce a 1 bp insertion or a MH-dependent 3 bp deletion, respectively (Fig. 2b). We used ssDNA (with homology arms) or dsDNA donor (without homology arms) to assess 34 bp insertions mediated by HDR or NHEJ[45]. KI-Seq data was classified into five DNA repair outcomes: two types of KI events based on 1. NHEJ; or 2. HDR (NHEJ-KI and HDR-

**Fig. 1 | High-throughput small molecule compounds screen identifies modulators of end joining (EJ) and homology-directed repair (HDR) DNA double-strand break (DSB) repair pathway choice. a** Outline of the small molecule screen in HEK293T cells harbouring a stable integrated traffic light reporter (TLR) construct in the *AAVS1* safe harbour locus. The reporter consists of a mutated enhanced green fluorescence protein (eGFPm) followed by out-of-frame T2A and *Discosoma sp.* red fluorescent protein (DsRed) sequences. Reading frames are specified based on the start codon position in eGFP. Flow cytometry analysis quantifies DSB repair events. DsRed fluorescence represents EJ events, while eGFP fluorescence depicts HDR. CMV cytomegalovirus promoter. **b** Illustration of the screening cascade that

identified DNA-PK inhibitors efficiently enhance HDR and reduce EJ repair events. pXC$_{50}$ value signifies the negative logarithm activity value of in vitro measured IC$_{50}$s, EC$_{50}$s, Ki, Kd, or percent inhibition endpoints. **c** Dot-plot illustrating the outcome of the primary screen assessing 20,548 compounds at 2 μM ($n = 1$). Only wells with cell counts above 1000 are illustrated. DMSO-treated cells serve as negative control, and 1 μM KU0060648 treatment as positive control. Compounds highlighted in black were selected for knock-in sequencing (KI-Seq) analysis. Dashed lines indicate set thresholds to categorise differences in EJ and HDR frequencies compared to DMSO control. Source data are provided as a Source Data file.

KI, respectively) and three different InDel outcomes associated with different repair events (Fig. 2a, b and Supplementary Fig. S2); 3. NHEJ, considering ±1 bp insertions and deletions; 4. MH-mediated deletions, considering deletion with flanking microhomologies ≥2 bp; and 5. other InDels (Fig. 2a, c). We utilised MH-associated deletions as a substitute to evaluate alt-EJ/TMEJ, disregarding other alt-EJ events such as synthesis-dependent alt-EJ[46]. Additionally, it should be noted that MH-mediated deletion can also result from NHEJ repair[47]. To further validate our pipeline, we used the potent DNA-PK inhibitor AZD7648[48] and showed improved CRISPR/Cas9-mediated HDR-KI upon compound treatment (Fig. 2c). Moreover, we confirmed that dsDNA integration (NHEJ-KI) and ±1 bp InDels resulting from NHEJ-mediated repair are inhibited by AZD7648 treatment. We also used KI-Seq to demonstrate that *Sp*Cas9-mediated cleavage can be used to promote directional NHEJ-mediated KIs, using predicted *Sp*Cas9 staggered cuts[34,49,50]. To do this, we designed dsDNA with 1 bp 5′-overhangs complementary to the target site. dsDNA templates with overhangs showed NHEJ-dependent directional insertions, as they were completely prevented by DNA-PK inhibition (Supplementary Figure S3d). Taken together, our results indicate that KI-Seq provides a robust pipeline that enables streamlined evaluation of DSB repair pathways and diverse KI strategies.

Next, we selected nine compounds from the TLR screen (Fig. 1c) to assess their effect on InDel profiles and their ability to increase HDR or NHEJ-mediated KIs using KI-Seq (Fig. 2d, e and Supplementary Fig. S4a–d). The compound set included three potent DNA-PK inhibitors (TLR1, AZD7648 and M9831/VX-987) and inhibitors of several DNA repair proteins, such as ATM (AZ31)[51], ATR (AZ20)[52], FEN1 (FEN1-IN-1)[53] and PARP1 (AZ2281)[54], or cell cycle effectors such as Aurora B Kinase (AURBK) (AZD2811)[55]. Additionally, we tested a LIG4 inhibitor (SCR7) which was not included in the TLR screen but has been shown to increase HDR[56]. All tested DNA-PK inhibitors increased ssDNA integration in a dose-dependent manner with a maximum of 2.9-fold increase for 1.25–3 μM AZD7648 (Fig. 2d and Supplementary Fig. S4c). Conversely, NHEJ-mediated integrations were almost completely inhibited (Fig. 2e and Supplementary Fig. S4d). In addition, treatment with DNA-PK inhibitors also led to a dose-dependent decrease of ±1 bp InDels and a concomitant increase of deletions associated with MH (Supplementary Fig. S4a, b). Furthermore, we compared these three DNA-PK inhibitors with Nu7026 and M3814, two additional DNA-PK inhibitors (Supplementary Fig. S3b, c). TLR1, M3814, M9831/VX-984 and AZD7648 belong to a newer generation of DNA-PK inhibitors, characterised by higher selectivity for DNA-PK over other phosphoinoside 3-kinases (PI3Ks). Both Nu7026 and M3814 have previously been shown to enhance Cas9-mediated integrations[27–29,57–60]. In contrast to the DNA-PK inhibitors identified from the TLR screen, Nu7026 showed no significant improvement for ssDNA integration and only partially inhibited dsDNA integration (Supplementary Fig. S3b, c). Unlike other tested DNA-PK inhibitors, treatment with 3 μM M3814 exhibited signs of toxicity, as implied by a reduction of cell confluency and overall poor editing efficiency. ATM and ATR inhibitors decreased ssDNA-mediated KI dose-dependent (Fig. 2d and Supplementary Fig. S4c). ATM inhibitor AZ31 increased the fraction of 1 bp duplications with a 3.2-fold increase compared to DMSO-treated samples, which is in agreement with a

previous report using other ATM inhibitors (Supplementary Fig. S4a, b)[24]. FEN1 and PARP1 inhibitors did not alter the distribution of DNA repair outcomes, although both enzymes were described to be involved in alt-EJ repair[14,61,62]. Inhibition of AURBK increased total editing efficiency to more than 90% without altering relative frequencies. This could potentially be linked to AURBK's critical role in checkpoint regulation[63]. The LIG4 inhibitor SCR7 showed no improvement of HDR compared to DMSO-treated controls (Supplementary Fig. S3b, c). In summary, we confirmed that DNA-PK inhibitors are strong promoters of HDR events, with AZD7648 exhibiting high potency and selectivity[48].

## AZD7648 efficiently promotes precise integration by inhibiting NHEJ events in dividing cell lines

DNA repair outcomes are dependent on cell background as well as cell cycle status[64]. We, therefore, investigated the effect of DNA-PK inhibition on editing outcomes in various transformed (HEK293T, Jurkat and HepG2) and non-transformed cells (human induced pluripotent stem cells (hiPSCs), primary human CD4 + T cells and primary human hepatocytes (PHH)) (Fig. 3). We delivered ribonucleoprotein (RNP) complexes of *Sp*Cas9 and gDel, gIns or gMej with or without donor DNA via electroporation. The cells were treated with either AZD7648 or DMSO immediately following electroporation (Fig. 3a). We detected differences in mutational signatures at the analysed loci in different cell lines, although the most representative variant for each target site was reproducible across different cell lines in DMSO-treated cells (Supplementary Fig. S5). The most frequent template-independent insertions resulted from NHEJ with duplications of one nucleotide proximal to the cut site, being the most abundant event in all cell lines except in Jurkat cells, where we noticed, as previously reported[65], peculiar 1–2 bp insertions of guanines or cytosines across all target sites. HepG2 and PHH displayed high levels of NHEJ-mediated repair as indicated by elevated relative frequencies of ±1 bp InDels. All cell lines showed MH-associated deletions at all target sites. These deletions were most frequent in HEK293T and hiPSCs. DNA-PK inhibition reduced the number of InDels in HepG2, Jurkat, primary human T cells and PHH and led to longer deletions with MH. HEK293T and hiPSCs fully compensated the inhibition of NHEJ InDels with increased MH-associated deletions (Supplementary Fig. S6). Next, we evaluated the KI profiles (Fig. 3b–g). NHEJ-mediated KIs were present in all tested cell lines (Fig. 3, NHEJ-KI). Conversely, HDR-mediated KIs were absent in PHH, demonstrating the strict dependency of HDR on cell cycle[66]. HDR-mediated integration efficiencies in untreated cells were well below 10% in most cell lines across all tested loci (Fig. 3, HDR-KI). Treatment with AZD7648 resulted in significant gains in KI efficiency across all cell lines except for PHH. For example, treatment with 1 μM AZD7648 increased HDR-mediated KI up to 4.9-fold in HEK293T, 5.2-fold in Jurkat, 6.1-fold in HepG2, 3.6-fold in hiPCS and maximal 12.6-fold in primary human CD4+T cells. AZD7648 treatment consistently reduced NHEJ-dependent InDels by up to 13.7-fold in HEK293T, 4.3-fold in Jurkat, 3-fold in HepG2, 6.5-fold in hiPSC and 22.4-fold in primary human T cells. In summary, we demonstrated that AZD7648, a potent and well-tolerated DNA-PK inhibitor, increases HDR-mediated integrations and reduces NHEJ-mediated deletions across different cell lines and genomic loci.

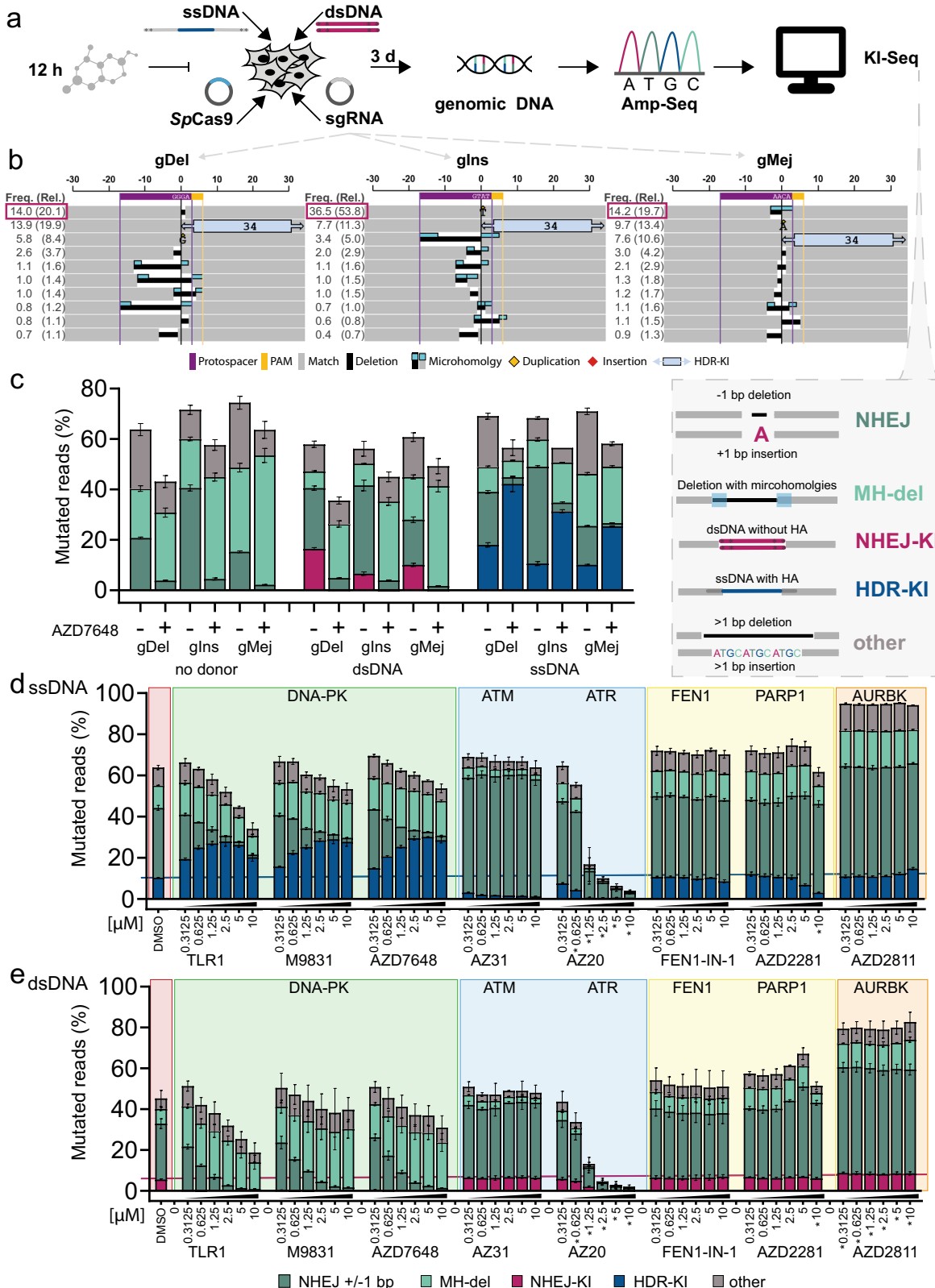

## Simultaneous targeting of DNA-PK and POLQ increases precise genome editing

Considering that DNA-PK inhibitor treatment increases the frequency of long deletions with MH usage, we hypothesised that inhibition of enzymes involved in TMEJ repair could further improve the frequency of desirable gene editing outcomes. Accordingly, we used knock-out cell pools featuring deletions of genes postulated as TMEJ mediators (*PARP1*,

*LIG3* or *POLQ*) to identify potential targets suitable for a co-inhibition strategy. We analysed InDel profiles at *Sp*Cas9-edited gDel, gIns and gMej loci with and without dsDNA/ssDNA in HEK293T knock-out pools treated with AZD7648 or DMSO. TIDE analysis[67] showed knock-out efficiencies of 94% (*PARP1*-KO pool), 92% (*POLQ*-KO pool) and 88% (*LIG3*-KO pool). No pronounced change in InDel profiles for the *PARP1*-KO cell pool was observed alone or in combination with AZD7648

**Fig. 2 | Knock-in sequencing (KI-Seq) determines DNA repair outcomes at *Sp*Cas9-induced DNA double-strand breaks (DSBs). a** Schematic of plasmid-based genome editing strategy to study effects of small molecules on DNA repair patterns in HEK293T cells. Deep-targeted amplicon sequencing data is analysed with KI-Seq. Alterations to the reference DNA sequence are assigned to different DNA DSB repair pathways: ±1 bp insertions/deletions = non-homologous end joining (NHEJ), deletions with microhomologies (MH-del), i.e., ≥ 2 bp = alternative end joining (alt-EJ), integration of double-stranded DNA (dsDNA) donor without homology arms = NHEJ-KI, integration of a single-stranded DNA (ssDNA) donor with homology arms = HDR-KI, all remaining events are summarised in "other". **b** Representative InDel profiles of ssDNA integration in HEK293T cells visualised by KI-Seq for target sites selected for their main mutation. gDel: 1 bp deletion, targeting *STAT1*, gIns: 1 bp insertion and gMej: 3 bp deletion associated to microhomologies, both targeting *CD34*. The graphical representation of the top ten variants shows ±30 bp around the *Sp*Cas9 cut-site. Each variant is annotated with absolute frequencies (Freq.) = fraction of mapped reads and relative frequencies (Rel.) in brackets = fraction of mutated reads. PAM protospacer adjacent motif. **c** Distribution of different repair events at the selected target sites, without DNA donor, with dsDNA or ssDNA donor, analysed with KI-Seq. Each transfection condition includes treatment with 1 μM DNA-PK inhibitor AZD7648 or corresponding DMSO control. Bar graphs represent mean values ± standard deviation (*n* = 3, technical replicates). **d, e** Mean fraction of mutated reads ± standard deviation (*n* = 3, technical replicates) by repair pathways. HEK293T cells were treated with different concentrations (0.3–10 μM) of DNA repair inhibitors from the traffic light reporter (TLR) screen or DMSO control and transfected with *Sp*Cas9, gIns and (**d**) ssDNA or (**e**) dsDNA donor. The horizontal line denotes the mean knock-in efficiency in DMSO-treated cells. Asterisk indicates treatments affecting cell confluency. Compounds and their associated targets are shown. Source data are provided as a Source Data file.

treatment (Fig. 4a–c and Supplementary Fig. S7a, b). The result confirms our PARP1 inhibitor data (Figs. 1c and 2d, e) and is in accordance with a previous study performed in mouse embryonic stem cells[33]. In contrast, we observed an increase in ssDNA integration in *LIG3*-KO and *POLQ*-KO pools on all targeted sites in combination with AZD7648 when compared to wild-type cells (Fig. 4c and Supplementary Fig. S7a, b). We noticed a more pronounced KI efficiency improvement in AZD7648-treated *POLQ*-KO compared to *LIG3*-KO cell pools at all tested target sites (Fig. 4c and Supplementary Fig. S7a, b). Remarkably, we detected a strong decrease of InDels in the AZD7648-treated *POLQ*-KO cell pool, leading to a high number of unmodified alleles in cells transfected with no donor (Fig. 4a and Supplementary Fig. S7a, b) or dsDNA (Fig. 4b and Supplementary Fig. S7a, b) and almost pure KIs in cells transfected with ssDNA (Fig. 4c, d and Supplementary Fig. S7a, b). Since we only observed a slight increase of ssDNA integrations in AZD7648-treated *LIG3*-KO pools and considering the essential role of LIG3 in mitochondrial DNA repair and cell viability[68], we decided to focus our efforts on PolΘ as the primary target for our dual inhibition strategy.

## Potent PolΘ inhibitors combined with AZD7648 increase HDR-mediated integrations and editing precision

A recent study reported that novobiocin, a natural antibiotic that targets bacterial topoisomerases as well as ATPases such as Hsp90, also inhibits PolΘ[39]. Additionally, novobiocin has previously been used to synergistically increase biallelic KI in combination with the DNA-PK inhibitor M3814 in mouse embryonic stem cells[69]. Therefore, we evaluated the effect of various novobiocin concentrations in combination with AZD7648 (1 μM) treatment on ssDNA integration (Supplementary Fig. S8a). Similar to the previous study[69], we detected only minor effects on KI efficiency in double-treated cells compared to AZD7648 treatment alone, an observation that contrasts with results obtained in our *POLQ*-KO pool. The discrepancy between pharmacological inhibition and genetic KO studies might be due to the promiscuous nature of novobiocin. Therefore, we initiated a literature search for a more selective and potent PolΘ inhibitor and identified a recently reported compound, ART558[37]. We also synthesised two additional disclosed compounds that we named PolQi1 (WO2021/028643) and PolQi2 (WO202/0243459). ART558 and PolQi1 target the polymerase domain of the enzyme, while PolQi2 inhibits the helicase activity of PolΘ. Supplementary Figure S8b depicts the structure of PolQi1 and PolQi2, along with the biochemical data reported in their published patents. To validate the inhibitors and investigate their efficacy on cellular TMEJ repair, we pre-treated HEK293T cells with different concentrations of PolQi1 or PolQi2. We assessed the mutational profile at three different target sites and found that cells treated with PolQi1 or PolQi2 showed a partial reduction of MH-associated deletion, similar to our *POLQ* knock-out pool data (Fig. 4a and Supplementary Figs. S7a, b and S8c, d). Lastly, we compared the dose-response curves and IC50 values of PolQi1 and PolQi2 to ART558 in the presence of AZD7648. The IC50

values were calculated based on the response of MH-mediated deletion. We observed similar potency between PolQi1 and ART558, while PolQi2 displayed approximately ten times higher potency (Supplementary Fig. S8e). Next, we tested these PolΘ inhibitors for their ability to increase ssDNA-mediated KIs and to reduce InDels in a dose-dependent manner at the gMej target site in HEK293T cells in combination with AZD7648 (Supplementary Fig. S8f). ART558 and PolQi1 showed a similar dose-depended increase of KI efficiencies with maximal improvement at a 3 μM concentration, with PolQi1 exhibiting a better HDR to InDel ratio at 3 and 10 μM (7.7:1 and 23.3:1) compared to ART558 (5.9:1 and 5.4:1). Interestingly, even at concentrations as low as 0.3 μM, PolQi2 achieved the same amount of KI and an HDR to InDel ratio of 10:1, and this was further improved with increasing concentrations of the compound. Next, we tested the compound combination of AZD7648 and 3 μM PolQi1 or PolQi2 at gMej, gDel and gIns target loci in HEK293T and Cas9-inducible hiPSCs (Fig. 4e, f and Supplementary Fig. S8h, i). In AZD7648-treated hiPSCs, we measured up to a 6.6-fold increase of ssDNA integration for PolQi1 and an 11.3-fold increase for PolQi2 inhibitor combination, while InDels were decreased up to 2.3- and 4.9-fold, respectively. In HEK293T cells, we observed a 3.9-fold increase of HDR-mediated integration for both PolΘ inhibitor combinations and a 17.4-fold and 56.3-fold reduction of InDels for PolQi1 and PolQi2, respectively. Additionally, we investigated the effect of combination treatments on cells transfected without a DNA donor template and observed up to a 12.3-fold decrease of InDels in AZD7648+PolQi1 and a 31.6-fold decrease in AZD7648+PolQi2 treated cells (Supplementary Fig. S8g).

Based on the above observations, we reasoned that dual compound treatment might work similarly to reduce InDels at off-target sites in the absence of the complementary donor sequences. To test this hypothesis, we selected two spacer sequences with well-known off-target sites, *HEK3* and *HEK4*[70]. Then, we delivered *Sp*Cas9 and sgRNA plasmids with and without ssDNA in the presence of either AZD7648 alone or in combination with PolQi1 or PolQi2 for evaluation of on- and off-target (seven off-target sites) gene editing activity in HEK293T cells. Combination treatment with AZD7648 and PolΘ inhibitors reduced both on-target and off-target InDels at all analysed loci compared to DMSO or AZD7648-treated cells (Fig. 5a).

Previous studies have shown that alt-EJ repair is associated with translocations and large deletions spanning kilobases around the Cas9 cut site[33,71,72]. Furthermore, NHEJ inhibition has been shown to increase, while TMEJ deficiency decreases, the accumulation of large deletions[31,33]. In line with this, we observed a reduction of large deletions in cells treated with either PolQi1 or PolQi2 and that adding a DNA donor decreased large deletion frequencies in DMSO-treated cells (Fig. 5b and Supplementary Fig. S9a, b). Treatment with AZD7648 increased the occurrence of large deletion in cells transfected with ssDNA or HDR plasmid donor compared to controls, an effect that was reversed by combination with PolΘ inhibitors. Furthermore, we

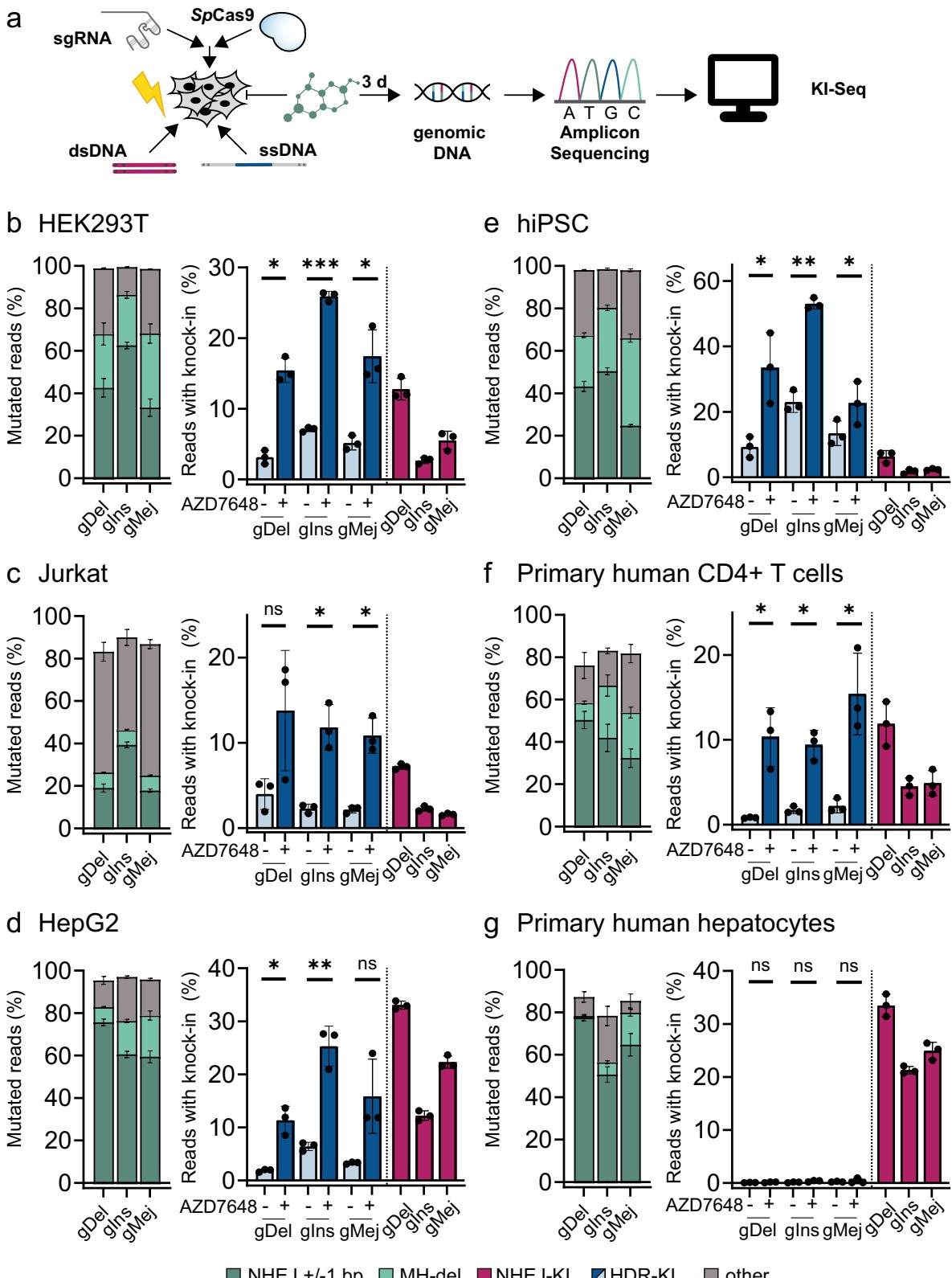

observed that combined inhibition of DNA-PK and PolΘ resulted in cells with fewer balanced translocations when simultaneously edited at two genomic loci (Fig. 5b, c and Supplementary Fig. S9c).

Taken together, our results demonstrated that combining AZD7648 with PolQi1 or PolQi2 improves the precision of gene editing at various loci and in different cell lines. Moreover, simultaneous inhibition of NHEJ and TMEJ significantly decreases Cas9-associated

off-target effects. Hereafter, we refer to the combined DNA-PK and PolΘ inhibition strategy as 2iHDR.

**2iHDR improves targeted KI in various cell types and with different HDR donor templates**

To evaluate the versatility of 2iHDR, we tested the strategy across different loci, cell lines, donor templates and integration-to-cut-site

**Fig. 3 | Characterisation of cell-line-specific *Sp*Cas9-mediated DNA double-strand break (DSB) repair with KI-Seq. AZD7648 promotes efficient integration of short single-stranded DNA donors in dividing cells. a** Workflow of knock-in sequencing (KI-Seq) using ribonucleoprotein (RNP)-based genome editing to study DSB repair and different knock-in strategies in various cell lines in the presence of 1 μM DNA-PK inhibitor AZD7648 or DMSO control. **b**–**g** Deep targeted amplicon sequencing data is analysed with KI-Seq. Alterations to the reference DNA sequence are assigned to different DNA DSB repair pathways: ±1 bp insertions/deletions = non-homologous end joining (NHEJ), deletions with microhomologies (MH-del), i.e., ≥ 2 bp = alternative end joining (alt-EJ), integration of double-stranded DNA (dsDNA) donor without homology arms = NHEJ-KI, integration of a single-stranded DNA (ssDNA) donor with homology arms = HDR-KI, all remaining events are summarised in "other". Left: fraction of mutated reads of different repair events at depicted target sites without donor or compound treatment. Right: knock-in

efficiencies of dsDNA or ssDNA in the presence of 1 μM AZD7648 or DMSO for three immortalised (**b**) HEK293T, (**c**) Jurkat, (**d**) HepG2, and three primary cell lines (**e**) *Sp*Cas9-inducible human induced pluripotent stem cells (hiPSC), (**f**) primary human CD4 + T cells and (**g**) primary human hepatocytes (PHH). Bar graphs represent mean values ± standard deviation (*n* = 3, biological replicates) calculated as percent of mapped reads. Significance level of ssDNA-mediated knock-ins was evaluated using Student's paired *t* test (two-tailed) *P < 0.05, **P < 0.01, ***P < 0.001. Calculated *P* values: DMSO vs AZD7648 (HEK293T gDel = 0.0152, gIns = 0.0001, gMej = 0.0183; Jurkat gDel = 0.0852, gIns = 0.0178, gMej = 0.0141; HepG2 gDel = 0.0203, gIns = 0.0084, gMej = 0.087; hiPSC gDel = 0.0316, gIns = 0.0077, gMej = 0.0365; primary human CD4 + T cells gDel = 0.0417, gIns = 0.0121, gMej = 0.0294; PHH gDel = 0.2543, gIns = 0.1686, gMej = 0.6828). Source data are provided as a Source Data file.

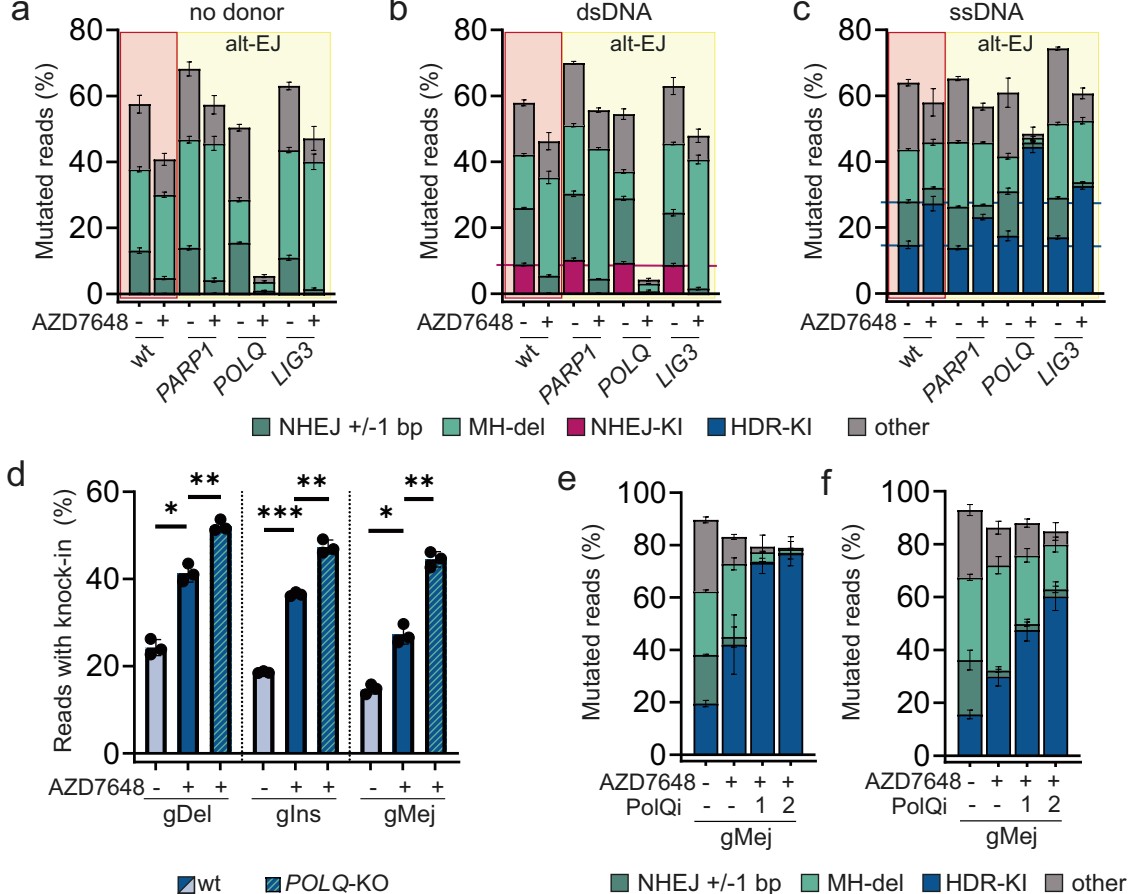

**Fig. 4 | Simultaneous Polθ and DNA-PK inhibition increases the frequency and precision of single-stranded DNA-mediated integration. a**–**c** Fraction of mutated reads of different repair events (±1 bp insertions/deletions = non-homologous end joining (NHEJ), deletions with microhomologies (MH-del), i.e., ≥2 bp = alternative end joining (alt-EJ), integration of double-stranded (dsDNA) donor without homology arms = NHEJ-KI, integration of a single-stranded (ssDNA) donor with homology arms = HDR-KI, all remaining events are summarised in "other") at the gMej target site with indicated donor DNA types for HEK293T wild-type (wt) cells and three knock-out pools of enzymes involved in alt-EJ repair. TIDE analysis estimated knockout efficiencies of cell pools were *PARP1* = 94%, *POLQ* = 92% and *LIG3* = 88%. Experiments were performed with 1 μM AZD7648 or DMSO control treatment 3 h before plasmid transfections. Horizontal lines illustrate the mean knock-in efficiency in DMSO or AZD7648-treated wt cells. Bar graphs show mean values ± standard deviation (*n* = 3, technical replicates). **d** Knock-in efficiencies of ssDNA donor integration at selected target sites with 1 μM AZD7648 or DMSO for

HEK293T wt and *POLQ*-KO cell pools. Bar graphs represent mean values ± standard deviation (*n* = 3, technical replicates). Statistical differences were evaluated using Student's paired *t* test (two-tailed) *P < 0.05, **P < 0.01, ***P < 0.001. Calculated *P* values: wt DMSO vs wt AZD7648 (gDel = 0.0135, gIns = 0.0003, gMej = 0.0143); wt AZD7648 vs *POLQ* KO AZD7648 (gDel = 0.0037 gIns = 0.0037 gMej = 0.0058). **e** Frequencies of different repair events at the gMej target site in plasmid and ssDNA transfected HEK293T cells. Cells were treated 1–3 h before transfections with DMSO,1 μM AZD7648, and 1 μM AZD7648 in combination with 3 μM Polθ inhibitor PolQi1 or PolQi2. Bar graphs represent mean values ± standard deviation (*n* = 3, biological replicates). **f** Percentage of different repair events in *Sp*Cas9-inducible hiPSC transfected with sgRNA gMej in the presence of 1 μM DNA-PK inhibitor AZD7648 or 1 μM AZD7648 in combination with 3 μM PolQi1 or PolQi2. Bar graphs depict the mean percentage of mutated reads ±standard deviation (*n* = 5, biological replicates). Source data are provided as a Source Data file.

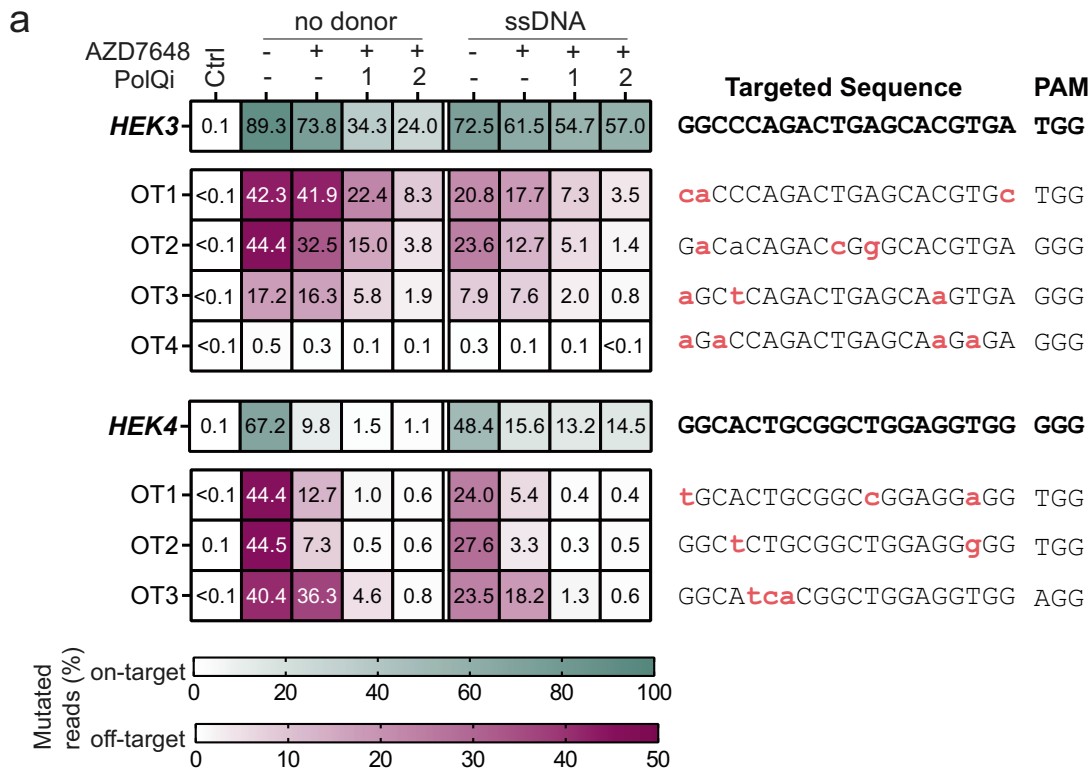

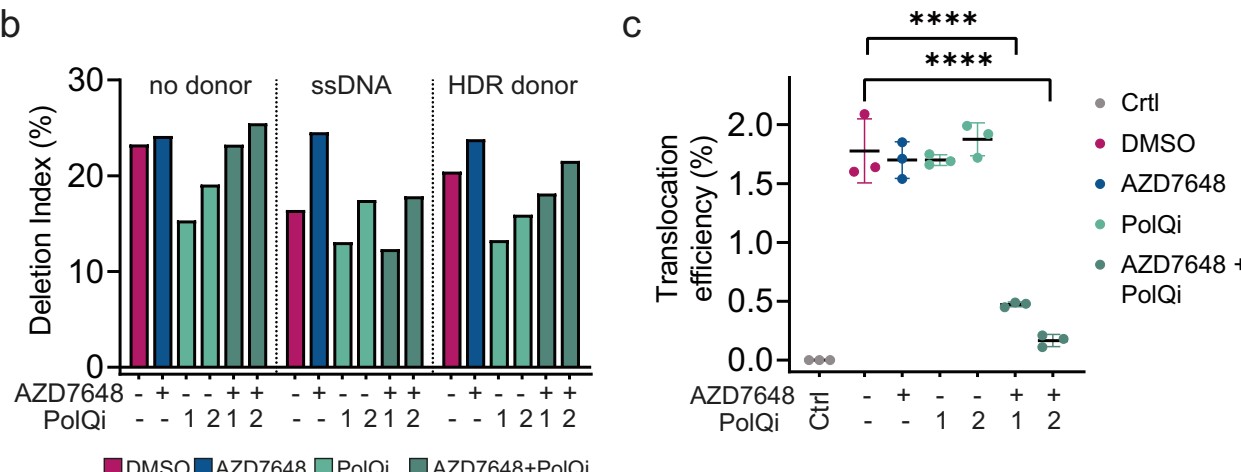

**Fig. 5 | 2iHDR treatment reduces off-target mutations and translocations.**
**a** Heatmaps show mean fraction of mutated reads (InDel frequencies) for *HEK3* and
*HEK4* on- and off-target sites (*n* = 3, technical replicates). Cells were treated with
DMSO, 1 μM AZD7648 and 1 μM AZD7648 with 3 μM PolQi1 or PolQi2. Ctrl refers to a
non-targeting control guide. Divergent bases are indicated on the right (small red
letters). **b** Quantification of large deletions from long-read sequencing data of
*HBEGF*-edited HEK293T cells treated with concentrations 1 μM AZD7648, 3 μM PolQi1,
3 μM PolQi2 and specified DNA donors.
Bar graphs present the percentage of deletion index[31] (*n* = 1). ssDNA: single-
stranded DNA donor with homology arms; HDR donor: plasmid with long

homology arms. **c** Simultaneous genome editing in HEK293T cells at *HBEGF* and
*PCSK9* loci for translocation analysis in the presence of inhibitors (1 μM AZD7648,
3 μM PolQi1 and 3 μM PolQi2) or DMSO control. Ctrl refers to a non-targeting
control guide. Dot-plots depict mean translocation frequencies analysed with
ddPCR ±standard deviation (*n* = 3, technical replicates). Statistical differences were
evaluated using one-way ANOVA (with Dunnett's post-hoc correction), *$P < 0.05$,
**$P < 0.01$, ***$P < 0.001$, ****$P < 0.001$. Calculated $P$ values: AZD7648 = 0.9424,
PolQi1 = 0.9424, PolQi2 = 0.8571, AZD7648 + PolQi1 = <0.0001, AZD7648 +
PolQi2 = <0.0001. Source data are provided as a Source Data file.

distances. First, we evaluated the effects of 2iHDR in HeLa and Jurkat
cells using endogenous gene targeting by integrating the small HiBiT
(39 bp) tag[73] in five different genes. After three days, we confirmed the
integration and expression of HiBiT by droplet digital PCR (ddPCR)
and luminescence detection (Fig. 6a, b and Supplementary Fig. S10a,
b). Indeed, 2iHDR treatment improved editing outcomes in all cases
tested, compared to untreated controls. We observed up to a 7.4-fold

increase of HiBiT integration and a 5.8-fold increase of expression in
HeLa cells. The benefits were even more striking in Jurkat cells, with a
maximal increase of 14.9-fold of HiBiT integration and an 8.7-fold
elevation in luminescence assay compared to untreated cells. In 9 out
of 10 conditions, treatment with 2iHDR using either one of the PolΘ
inhibitors exhibited improved KI efficiencies over AZD7648 treatment
alone. Even when integration levels were not increased compared to

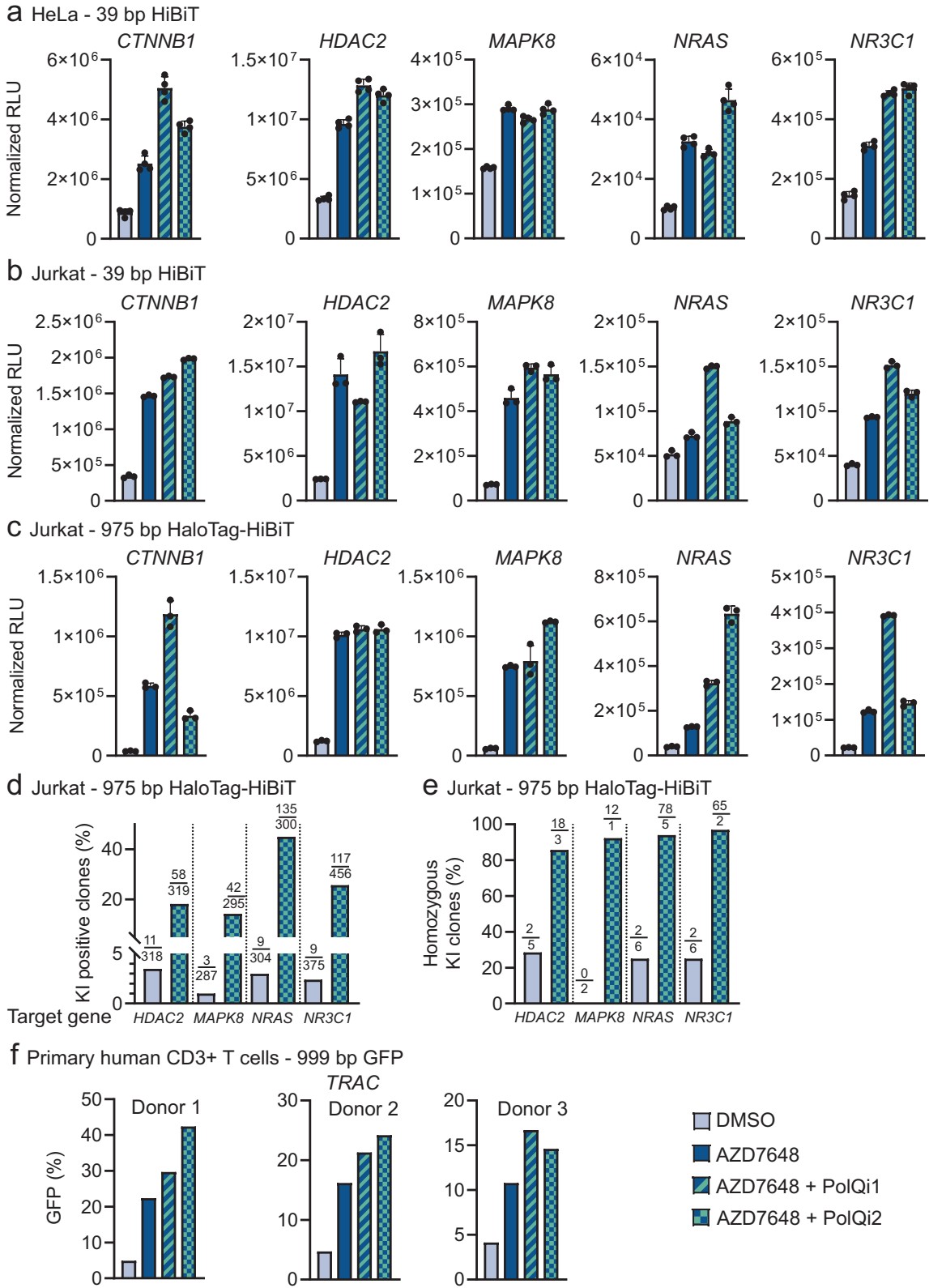

AZD7648 alone, 2iHDR consistently reduced InDel formation at on-target sites (Supplementary Fig. S11a, b).

To confirm that 2iHDR can also be used with double-stranded donor templates and larger donor integrations, we performed KIs of the 975 bp HaloTag-HiBiT fusion tag into five loci in Jurkat cells. We analysed the cells for HaloTag-HiBiT insertions 14 days after transfection using ddPCR and luminescence detection of HiBiT. Dual

compound treatment increased KI efficiencies up to 31-fold (luminescence) or 25.7-fold (ddPCR) compared to untreated cells (Fig. 6c and Supplementary Fig. S10c). At four targeted sites, 2iHDR incorporating either one of the PolΘ inhibitors showed higher integrations than AZD7648 treatment alone.

Notably, integration efficiencies were detected after 2 weeks, which indicates that KIs produced by 2iHDR are stable over time and

**Fig. 6 | 2iHDR boosts integration of different DNA donor templates with various integration-to-cut-site distances in diverse cell lines. a–c** Quantification of luminescence signal via HiBiT lytic assay to estimate protein tagging efficiencies with HiBiT and Halo-HiBiT tags at indicated genomic loci and cell lines. Experiments were performed with 1 μM AZD7648 and 1 μM AZD7648 in combination with 3 μM PolQi1 or PolQi2. Data represent mean luminescence signal after background subtraction ±standard deviation (Hela HiBiT, $n = 4$; Jurkat HiBiT, $n = 3$; Jurkat HaloTag-HiBiT, $n = 3$; all technical replicates). RLU relative light unit **d** Bar graphs represent the percentage of Jurkat cell clones containing HaloTag-HiBiT knock-in assessed with the Nano-Glo HiBiT lytic detection system for the indicated target genes. Cells were treated with DMSO or 1 μM AZD7648 and 3 μM PolQi2. Numbers show knock-in (KI) positive clones per total analysed clones. **e** ddPCR analysis of HaloTag-HiBiT KI-positive Jurkat cell clones. Bar graphs show the percentage of homozygous HaloTag-HiBiT integration at different loci using DMSO or 1 μM AZD7648 and 3 μM PolQi2. Numbers represent homozygous per heterozygous clones. **f** Flow cytometry analysis to measure GFP integration into the *TRAC* locus in primary human CD3 + T cells derived from three individual human donors in the presence of 1 μM AZD7648, and 1 μM AZD7648 in combination with 3 μM PolQi1 or PolQi2. Knock-in efficiencies were evaluated 7 days post-electroporation. Data represent the percentage of GFP-positive cells in the viable cell population. Source data are provided as a Source Data file.

that cells were still able to proliferate after treatment. To confirm this, we performed HaloTag-HiBiT integration in Jurkat cells at four different loci, as described above, followed by single cell sorting and expansion for about three weeks. We measured 5.3-fold *HDAC2*, 14-fold *MAPK8*, 15-fold *NRAS* and 4.8-fold *NR3C1* increase of HiBiT positive clones in 2iHDR treated cells (Fig. 6d). Analysis with ddPCR revealed that more than 80% of the positive clones are homozygous for HaloTag-HiBiT integration (Fig. 6e). This result confirmed that 2iHDR treated cells could proliferate in the presence of stable, homozygous integration of large genetic payloads.

Next, we tested the performance of 2iHDR treatment in primary T cells by integrating a GFP expression cassette into the *TRAC* locus. To this end, we delivered RNPs and a GFP encoding dsDNA homology-directed repair template (HDRT) into T cells isolated from three healthy donors. KI efficiency in untreated cells was below 5%, whereas DNA-PK inhibition improved efficiency by 2.6- to 4.6-fold. Treatment with 2iHDR yielded additional improvements, with 2iHDR using AZD7648+PolQi1 showing a 4.0- to 6.0-fold increase and 2iHDR using AZD7648+PolQi2 showing a 3.5- to 8.6-fold increase (Fig. 6f). Furthermore, 2iHDR treatment did not significantly affect cell viability while increasing KI+ cell yields for two out of three T cell donors compared to DNA-PK inhibition alone (Supplementary Fig. S10d). Overall, our data show that 2iHDR treatment is a convenient and powerful method to improve the efficiency and precision of CRISPR-Cas9 genome editing with broad applicability across multiple genomic loci and in various cell lines and primary cell models.

## Discussion

The CRISPR-Cas system is a highly versatile tool to introduce targeted DNA lesions. Upon DNA cleavage, repair is accomplished through activation of intrinsic mechanisms, yielding target site-specific DNA modifications, such as deletions and insertions. However, the main obstacle to achieve precise genome editing is that HDR, the DNA repair pathway yielding precise editing, is not frequently used and is commonly outperformed by NHEJ and alt-EJ pathways, both resulting in the formation of InDels and imprecise edits. To address this problem, we conducted a high-throughput screening campaign to identify and validate compounds that enhance HDR while reducing undesirable side effects of Cas9 treatment. We focused on manipulation of the HDR pathway since this strategy has been previously established to improve the precision and efficiency of CRISPR-Cas9 genome engineering[74]. The identification of numerous DNA-PK inhibitors as enhancers of precision KI confirmed the key role of NHEJ in modulating HDR. We found that the new generation of DNA-PK inhibitors strongly decreases NHEJ and subsequently increases HDR, in line with their improved potency and selectivity[48,60]. Accordingly, we selected AZD7648 as the lead compound due to its strong enhancement of promoting KIs across different target sites in various cell backgrounds and its established safety record[48].

To further characterise the effects of AZD7648 and other small molecules that emerged from our screening campaign, we developed KI-Seq, an NGS-based pipeline that neither depends on a single predefined sgRNA nor requires previous transgene integration, therefore extending its utility to any cell type, including non-dividing primary cells. This allowed us to perform an unbiased analysis of InDel profiles and to determine the contribution of not only HDR but also of NHEJ and alt-EJ DNA repair following CRISPR-Cas9 treatment. Unlike most open-access analysis tools that separate Cas9-induced DNA cleavage events into homology-directed KIs and combine all other mutation types into a single category[43,75,76], KI-Seq provides a much more detailed analysis. For example, we confirmed that treatment with ATM inhibitors promotes +1 bp duplications[24] and observed that AURBK inhibitor stimulates an increase across all editing events. In addition, KI-Seq enables the analysis of exogenous dsDNA donor capture after DSB formation, which is of notable interest in HDR-deficient, non-dividing cells and underscores the key role of NHEJ repair in these cell types[23,77]. Using KI-Seq, we demonstrated that AZD7648 and other DNA-PK inhibitors increase ssDNA integration in a dose-dependent manner while inhibiting NHEJ-mediated integration and increasing deletions associated with micro-homologies. Therefore, from these studies, we conclude that AZD7648 is a potent DNA-PK inhibitor that increases ssDNA integration and reduces NHEJ-mediated deletion across different cell lines.

We next asked how inhibition of both NHEJ and alt-EJ affects editing outcomes. In fact, we and others have previously shown that alt-EJ plays a dominant role in InDel generation upon DNA-PK inhibition after Cas9-induced DSB[29–31]. These findings suggested that additional pharmacological inhibition of the alt-EJ repair pathway, including PolΘ, might lead to a further increase of HDR. Using genetic deletion, we confirmed that *POLQ*-KO combined with AZD7648 treatment results in improved HDR-mediated precision gene editing. In this study, we tested three recently reported inhibitors, which either targeted the polymerase (ART558 and PolQi1) or the helicase domains (PolQi2) of PolΘ. Notably, all three PolΘ inhibitors increased integration efficiencies in combination with AZD7648. These results led us to propose that combination treatment with both DNA-PK and PolΘ inhibitors (2iHDR) is a powerful strategy to dramatically improve the performance of precision KI gene editing. The 2iHDR strategy consistently performed better than untreated controls in increasing KI efficiencies independent of template sizes and donor formats, such as ssDNA or plasmids. We anticipate that 2iHDR could be combined with other non-viral[74] and viral HDR[78] templates to improve HDR-mediated genomic integrations. In addition, we observed an increase in precise integration to InDel ratios for short insertions across all tested target sites in various cell types with 2iHDR compared to AZD7648 alone. Another advantage of our 2iHDR approach is the reduction of InDel rates at target sites, therefore allowing secondary retargeting of the wild-type allele to further boost the editing efficiencies. Importantly, we also noticed a significant decrease in off-target editing when using 2iHDR, suggesting that dual inhibition of NHEJ and TMEJ could potentially improve the overall safety of CRISPR-Cas9 treatments[69]. Overall, these results indicate that the developed 2iHDR approach is superior to previously described strategies targeting only DNA-PK.

A current shortcoming of KI-Seq is the requirement of small amplicons for variant analysis, preventing the investigation of complex DNA rearrangements. To overcome this limitation, we applied ddPCR and long-read sequencing allowing for detailed analyses of large on-

target deletions and translocation events. In line with previously published data[41], we noticed that 2iHDR treatment does not affect Cas9-associated on-target large deletions and leads to a reduction of translocation events. These findings present an additional advantage of 2iHDR over the treatment with AZD7648 alone, where we detected an increase in large deletions. However, based on this study, we cannot exclude the possibility that 2iHDR has uncharacterised, undesirable side effects and further studies investigating the genome-wide impact of simultaneous NHEJ/MMEJ inhibition on the repair of spontaneously occurring DSBs are needed.

Going forward, we anticipate that both KI-Seq and 2iHDR will accelerate the use of CRISPR-mediated KI for research and clinical applications, given their broad applicability across a wide range of cell lines and significantly improved performance over existing strategies.

## Methods

### Ethical statement
AstraZeneca has a governance framework and processes in place to ensure that commercial sources have appropriate patient consent and ethical approval in place for collection of the samples for research purposes including use by for-profit companies. The AstraZeneca Biobank in the UK is licensed by the Human Tissue Authority (Licence No. 12109) and has National Research Ethics Service Committee (NREC) Approval as a Research Tissue Bank (RTB) (REC No 22/NW/0102) which covers the use of the samples for this project. NHS Blood & Transplant (NHS-BT UK) has provided material in support of the research. The views expressed in this publication are those of the author(s) and not necessarily those of NHS Blood & Transplant. Primary human T cells were derived from the blood of healthy donors recruited through AstraZeneca's voluntary blood donation programme, which was approved by AstraZeneca's institutional review board and local ethic committee (033-10). Informed consent was obtained for all human donors included in this study.

### Plasmids
Codon optimised SV-40 NLS-*Sp*Cas9-T2A-GFP was synthesised and cloned downstream of a cytomegalovirus (CMV) promoter. Spacer sequences for sgRNA transcription were cloned under the control of the U6 promoter containing the previously described sequence of the *Sp*Cas9 scaffold[79]. The spacer sequence was synthesised as DNA oligos with four bp 5′-overhangs complementary to the vector backbone. DNA oligos (Sigma-Aldrich or Integrated DNA Technologies) were annealed and ligated into *Aar*I (Thermo Fisher Scientific) linearised vector DNA using T4 DNA ligase (New England Biolabs) following the manufacturer's protocol. Plasmids were transformed in DH5-alpha or DH10-beta chemo-competent *Escherichia coli* (*E. coli*) cells (New England Biolabs). Transformed bacteria were grown in LB medium supplemented with 50 μg/mL kanamycin or 100 μg/mL carbenicillin. Plasmid and sgRNA sequences can be found in Supplementary Note 1 and Supplementary Data 1. The sequence fidelity of cloned constructs was confirmed through Sanger-sequencing (Genewiz/Azenta).

### Donor templates
ssDNA donor templates were ordered with two 5′- and 3′-terminal phosphorothioate linkages (Integrated DNA Technologies). Blunt-ended dsDNA donors for NHEJ-mediated integrations were generated by annealing two phosphorothioate end-modified oligos as described before[70]. HaloTag-HiBiT plasmid (pUCIDT-AMP) donor was synthesised by Integrated DNA Technologies. GFP dsDNA homology directed repair template (HDRT) were synthesised, cloned into vectors as required and sequence verified by external vendors (GenScript). HDRT was amplified from plasmid DNA (pAAV-SA-2A-GFP) by PCR using Phusion Flash High-Fidelity PCR Master Mix (Thermo Fisher Scientific) and primers anchored in the homology arms of the HDRT specific for KI at the human *TRAC* locus (Merck). PCR amplicons (typically 2.5 mL)

were purified using NucleoMag NGS Clean-up and Size Select (Macherey-Nagel) following the manufacturer's instructions and eluted in 450 μL distilled water. Subsequently, purified HDRT was concentrated using Amicon Ultra-2 Centrifugal Filter Units (Merck) according to the manufacturer's instructions, yielding typically 5 μg (25 μL) of HDRT. HDRTs were diluted as required for future experiments (typically 1.5 μg/μL) in distilled water. Purification and size of dsDNA HDRTs were confirmed by gel electrophoresis on a 1% agarose gel. All sequences are available in Supplementary Data 3.

### Synthetic guide RNAs
Modified synthetic sgRNAs were ordered from Agilent, Synthego or Integrated DNA Technologies. For HiBiT and HaloTag-HiBiT integrations, Alt-R CRISPR RNA (crRNA), Alt-R transactivating crRNA (tracrRNA) and nuclease-free duplex buffer were obtained from Integrated DNA Technologies. sgRNA was prepared by incubating 1 nmol crRNA, 1 nmol tracrRNA, and nuclease-free duplex buffer in a final volume of 50 μL at 95 °C for 5 min and then cooled to room temperature. Spacer sequences are summarised in Supplementary Data 1.

### Small molecules
AZ31, AZ20, AZD2281, AZD2811, AZD7648, TLR1 (WO2014183850), M9831/VX-984, FEN-IN-1, PolQi1 (WO2021/028643, Example 158), PolQi2 (WO2020/243459, Example 99) were provided by AstraZeneca (Gothenburg, SE). Three of the DNA-PK inhibitors, AZD7648 (HY-111783, MedChemExpress), M9831/VX-984 (HY-19939S, MedChemExpress) and M3814 (S8586, Selleckchem) are commercially available. Nu7026 and KU0060648 were purchased from Sigma-Aldrich (N1537, SML1257). Novobiocin, ART558 and AZD7648 were obtained from MedChemExpress (HY-B0425A, HY-141520 and HY-111783). SCR7 was purchased from Selleckchem (S7742). All compounds were dissolved in DMSO (Thermo Fisher Scientific) at a concentration of 10 mM.

### Cell culture
HEK293T cells (parental clone HEK-293, ATCC CRL-1573; Genhunter corporation Q401) and HEK293T-KO cell pools (PARP1-/-, LIG3 -/- POLQ-/-, Synthego, CRISPR KO pool) were maintained in DMEM (+GlutaMAX, High Glucose), with 10% fetal bovine serum (FBS). HepG2 (ATCC, HB-8065) were cultured in MEM (+GlutaMAX) with 1% (v/v) sodium pyruvate and 10% FBS. Jurkat cells (DKMZ, ACC-282) were maintained in RPMI1640 with 10% heat-inactivated FBS (all reagents from Gibco, Thermo Fisher Scientific). Jurkat cells for HiBiT and HaloTag-HiBiT experiments (TIB-152, American Type Culture Collection) were maintained in HEPES-Buffered RPMI (Gibco) supplemented with 10% FBS (Seradigm).

The hiPSC line (R-iPSC Clone J, LineID: iPS.1) was derived from foreskin-fibroblasts using the Stemgent mRNA Reprogramming Kit (Stemgent) as described previously[80] and maintained in the feeder-free culturing system Cellartis DEF-CS 500 (Takara). The inducible *Sp*Cas9 cell line was generated as described earlier[81].

Primary human T cells were derived from the blood of healthy donors recruited from the AstraZeneca voluntary blood donation programme. Peripheral blood mononuclear cells were isolated from fresh blood using Lymphoprep (STEMCELL Technologies) density gradient centrifugation. Total CD4 + T cells were enriched by negative selection with the EasySep Human CD4 + T Cell Enrichment Kit (STEMCELL Technologies) to an average purity of 90%. CD4 + T cells were propagated in RPMI-1640 medium containing the following supplements 1% (v/v) GlutaMAX-I, 1% (v/v) non-essential amino acids, 1 mM sodium pyruvate, 1% (v/v) L-glutamine, 50 U/mL penicillin and streptomycin and 10% heat-inactivated FBS (all from Gibco, Thermo Fisher Scientific). Activation of T cells was performed with the T Cell Activation/Expansion kit (Miltenyi). In all, $1 \times 10^6$ cells/mL were mixed with beads using a bead-to-cell ratio of 1:2, and $2 \times 10^5$ cells were seeded per well of a 96-well round-bottom tissue culture plate and

incubated for 24 h. The cells were pooled before electroporation and cultured in antibiotic-free medium for 24 h after electroporation.

Cryopreserved PHH were derived from a de-identified individual donor (F00995-P, Lot.: VNL, BioIVT). One vial of frozen PHH (-5 × 10^5 cells) was thawed in 50 mL of Cryopreserved Hepatocyte Recovery Medium (Gibco, Thermo Fisher Scientific). After nucleofection cells were seeded on collagen-coated 96-well cell culture plates (Corning, SigmaAldrich) containing culture medium (Williams' Medium E, 1% (v/v) L-glutamine-penicillin-streptomycin, 0.1 μM dexamethasone (Sigma-Aldrich), 1% (v/v) insulin-transferrin-selenium (Thermo Fisher Scientific)) and 10% FBS. In all, 2–4 h after nucleofection medium was changed to serum-free culture medium with 0.25 mg/mL Matrigel Basement Membrane Matrix (Corning, Sigma-Aldrich). Serum-free complete culture medium was changed daily. All cells were maintained at 37 °C in a 5% CO$_2$ atmosphere.

Primary human CD3 + T cells were isolated from a healthy donor Leukopaks (NHS-BT UK). Peripheral blood mononuclear cells (PBMCs) were isolated from whole blood by Ficol centrifugation using SepMate tubes (STEMCELL Technologies) and subsequent isolation of CD3 + T cells using negative selection isolation kits (STEMCELL Technologies) following the manufacturer's instructions. CD3 + T cells were cultured in TexMACS Medium (Miltenyi Biotech) supplemented with 5% heat-inactivated human AB serum (Merck), 100 units/mL penicillin (Merck) and 100 μg/mL streptomycin (Merck) and 300 IU/mL IL-2 (STEMCELL Technologies). T cells were seeded at 2 × 10^6 cells/mL (typically 10 × 10^6 cells per well of a six-well plate) and activated with ImmunoCult T Cell activator (STEMCELL Technologies) following manufacturer's instructions. 72 h post activation, editing was performed as described below. Cells were passaged, and media was changed as appropriate to maintain culture density as above. Sex and/or gender was not considered in the study design.

Cell lines were regularly tested for mycoplasma contaminations, and cell identity was confirmed through STR profiling.

### Generation of HEK293T traffic light reporter cell line
The traffic light reporter (TLR) construct was designed similar to Certo et al.[42] consisting of a defective enhanced green fluorescent protein (eGFP) mutated in amino acids 65-67 (eGFPm) and linked to out-of-frame T2A and DsRed under CMV promoter[42]. Sequences are provided in Supplementary Data 1. For TLR cell line generation, the TLR construct was stably integrated into the *AAVS1* locus of HEK293T cells using ObLiGaRe[21]. The ObLiGaRe-TLR construct was co-transfected with zinc-finger nuclease (ZFN)-AAVS1 plasmid in a 1:3 ratio using FuGENE transfection reagent (Promega) following the manufacturer's instructions. Transgene integration selection was performed with puromycin (Gibco) before single cell clones were generated by limited dilution. Clonal HEK293-TLR cells were validated for transgene integration in the *AAVS1* locus with junction PCR and copy number integration using ddPCR (Supplementary Data 2). Junction PCR was performed with Extensor Hi-Fidelity PCR Master Mix (Thermo Fisher Scientific) in a 20 μL reaction using 500 nM of target-specific primers and the following cycling conditions: 94 °C for 2 min, 35x (94 °C for 20 s, 55 °C for 20 s, 68 °C for 45 s) and 68 °C for 7 min. ddPCR set-up was as follows; the final ddPCR PCR reaction contained 1× ddPCR Supermix for Probes, no dUPT (Bio-Rad), 1× FAM-labelled puromycin assay (BioRAD), 1× AP3B1-HEX-labelled human reference assay (dHsaCP1000001, BioRAD), 1/40 *Hind*III (Thermo Fisher Scientific) and 50 ng genomic DNA. Automated droplet generation, PCR amplification, droplet reading, and analysis was performed as described below for the translocation assay. In brief, probes targeting puromycin were normalised to the *AP3B1* reference assay for copy number integration analysis. A clone with single-copy integration was selected and expanded. For the TLR screen, 1 × 10^8 HEK293-TLR cells were electroporated with 75 μg plasmid containing *Sp*Cas9 and eGFP-targeting sgRNA sequence, and 125 μg HDR donor

plasmids using MaxCyte STX (Maxcyte). Electroporated cell pools were cryopreserved until further use.

### TLR screen
In the primary TLR screen 20,548 compounds were tested at 2 μM ($n = 1$). For the screen, 5000 transiently transfected HEK293-TLR cells were seeded per well of a 384-well poly-D-lysine-coated cell culture plate (Greiner) containing the library compounds, DMSO (negative control), or 1 μM KU0060648 (positive control)[82]. The plates were incubated for three days at 37 °C before medium was removed and cells were detached with Trypsin-EDTA (0.25%) (Gibco). FACS buffer (PBS, pH 7.4 (Gibco) 1 mM EDTA (Ambion), 2% (v/v) FBS (Gibco) and FL-4 well identification beads (AH Diagnostic) were added to the wells and cells were transferred to 384-well round bottom plates (Corning). Samples were analysed with the Intellicyte iQue screener (Sartorius) and ForeCyt® Enterprise Client Edition 6.2 Software. The gating strategy is detailed in Supplementary Fig. S1b.

Samples eligible for the screen contained more than 1000 cells per well. Gates were defined based on control samples as follows; DsRed-positive cells >15% as an increase of EJ and DsRed-positive cells <8% as decrease of EJ. GFP-positive cells >4.5% were considered as an increase of HDR events. Compounds were regarded as inactive for DsRed-positive cells <15% and GFP-positive cells <4.5% (Fig. 1c).

To validate the results of the primary screen 380 compounds were re-analysed in a second hit confirmation screening round. To investigate if the effects of HDR and EJ were dose-dependent, all compounds were tested in a ten-point dilution series with dilution factor three ranging from 10 μM to 0.51 nM. Compound exclusion criteria were GFP/DsRed fluorescence values above 50% (false-positives) and lack of dose-dependency in the secondary screen. Compounds with random effects on a few points and no dose-response were considered inactive.

### Transfections – plasmid delivery
HEK293T cells were seeded into 96-well plates (15,000–25,000 cells/well), 48-well plates (45,000 cells/well) or 24-well plates (90,000 cells/well) 1 day prior to transfections. Compound treatment was initiated 1–4 hours before transfections with FuGENE HD transfection reagent (Promega) following the manual. In all, 50 ng of *Sp*Cas9, 30 ng of sgRNA were transfected with 1 pmol of dsDNA donor or 1–2 pmol of ssDNA donor per well of a 96-well plate. For large deletion analysis, 115 ng *Sp*Cas9, 70 ng of sgRNA (targeting *HBEGF* or non-targeting sgRNA) and 180 ng of HDR donor plasmid or 3 pmol of ssDNA were used per well of a 48-well plate. For the translocation assay, 375 ng *Sp*Cas9 and 120 ng of each sgRNA targeting *PCSK9* and *HBEGF* were transfected per well of a 24-well plate. Respective controls were set up with non-targeting sgRNAs.

### Transfections – sgRNA delivery to inducible Odin-SpCas9 hiPSC
ODinCas9-inducible hiPSC[81] were reverse transfected with sgRNAs using Lipofectamine RNAiMAX (Invitrogen, Thermo Fisher Scientific) according to the manual. In short, cells were sub-cultured 1 day before transfections. The next day, *Sp*Cas9 expression was induced with doxycycline (10 μg/mL) for 1 h. After 6 hours, cells were washed with PBS (Gibco, Thermo Fisher Scientific), detached with TrypLE Express Enzyme (no phenol red, Gibco, Thermo Fisher Scientific) and resuspended in cell culture medium. In all, 1.28 × 10^4 cells were mixed with 29 ng of sgRNA, 0.25 pmol of ssDNA or dsDNA, 0.3 μL RNAiMax in 5 μL OptiMEM (Gibco, Thermo Fisher Scientific). The cells were seeded into 96-well cell culture plates containing culture medium with DMSO or inhibitors.

### Transfections – RNP preparation and delivery
To prepare RNP complexes, Alt-R *Sp*Cas9 nuclease v3 (IDT) and sgRNA were incubated in a 1:1 molar ratio for 5–10 min at room temperature. RNPs were kept on ice until electroporation. Immediately before

transfections, DNA donors, electroporation enhancer oligo and cells were added to the RNPs. Prior to electroporation, cells were washed with PBS and cell number and viability were assessed using a Cedex HighRes (Roche) or Vi-cell XR (Beckman Coulter) cell counter. Cells were resuspended in respective electroporation/nucleofection buffer HEK293T cells and HepG2 in SF Buffer (SF Cell Line 4D-Nucleofector X Kit, Lonza), Jurkat cells in SE buffer (SE Cell Line 4D-Nucleofector X Kit, Lonza), PHH in P3 buffer (P3 Primary Cell 4D-Nucleofector X Kit, Lonza) and primary human CD4 + T cells in R buffer (Neon Transfection System). For electroporations, $2 \times 10^5$ HEK293T cells, $3.5 \times 10^5$ HepG2 cells and $2 \times 10^5$ PHH were combined with 100 pmol $Sp$Cas9, 100 pmol sgRNA, 40 pmol ssDNA and 60 pmol electroporation enhancer oligo, or 20 pmol dsDNA and 100 pmol electroporation enhancer oligo. In all, $5 \times 10^5$ Jurkat cells were electroporated with 25 pmol $Sp$Cas9, 25 pmol sgRNA, 25 pmol ssDNA or 25 pmol dsDNA and 25 pmol of electroporation enhancer oligo. In all, $5 \times 10^5$ primary human CD4 + T cells were electroporated with 30.5 pmol $Sp$Cas9, 36 pmol sgRNA, 25 pmol ssDNA or 25 pmol dsDNA, and 40 pmol electroporation enhancer. The following Lonza 4D-Nucleofector programs were used: HEK293T: CM-130, HepG2: EH-100, Jurkat: CL-120 and PHH: EX-147. Primary human CD4 + T cells were electroporated using the Neon system (Invitrogen) applying the following settings: 1600 V, width: 10 ms, and pulse number: 3. Afterwards, cells were distributed into six wells of a 96-well cell culture plate containing culture medium with DMSO or inhibitors.

### Integration of HiBiT and HaloTag-HiBiT tags

HiBiT integration at the different target sites required various cut-site-to-insert distances, depending on available protospacer sequences (integration distance to cut sites were: $CTNNB1$: 14 bp, $HDAC2$: 2 bp, $MAPK8$: 26 bp, $NRAS$: 10 bp, $NR3C1$: 12 bp). Whenever possible, we introduced additional silent mutations in the PAM. Prior to electroporation, cells were pre-treated for 24 h with combinations of 1 μM AZD7648, 3 μM PolQi1, or 3 μM PolQi2. RNP complexes were assembled by incubating 100 pmol $Sp$Cas9 and 120 pmol gRNA in a final volume of 10 μL nuclease-free duplex buffer for 10 min at ambient temperature. $6 \times 10^5$ cells were resuspended in 100 μL Ingenio Electroporation Solution (Mirus), and RNP complex with either 100 pmol ssODN or 7.5 μg plasmid donor DNA was added to the cell suspension. Cells containing RNP and donor DNA were electroporated at 150 V using the Ingenio EZporator Electroporation System (Mirus). Cells were incubated at ambient temperature for 5 min and transferred to a six-well plate containing 2 mL growth medium with or without combinations of 1 μM AZD7648, 3 μM PolQi1, or 3 μM PolQi2. HiBiT and HaloTag-HiBiT insertions were analysed 3- and 14-days post-electroporation, respectively.

### HiBiT detection assay

To screen for integration, $2 \times 10^4$ cells were plated in solid white 96-well tissue culture plates (Corning 3917) in 100 μL growth medium. HiBiT was detected using the Nano-Glo HiBiT Lytic Detection System (Promega) following the manufacturer's instructions. Briefly, 100 μL of Nano-Glo HiBiT Lytic Detection Reagent was added directly to the cells and incubated for 5 min on an orbital shaker (300 rpm) before recording luminescence on a GloMax Discover (Promega) with 0.2 s integration time.

### Clonal isolation of HaloTag-HiBiT transfected Jurkat cells

Pools of edited cells were resuspended to $5 \times 10^6$ cells per mL in sorting buffer (DPBS (Gibco, Thermo Fisher Scientific), 10 mM HEPES (Gibco, Thermo Fisher Scientific), 0.2% FBS, and 10 units per mL of penicillin-streptomycin (Gibco, Thermo Fisher Scientific), passed through a 35 μm mesh filter (Corning, Falcon) to disperse clumps, and loaded onto the BD FACSMelody (BD Biosciences) cell sorter. Single cells were sorted into each well of solid white 96-well tissue culture plates

(4–5 plates) (Corning) containing 150 μL growth medium per well. Cells were grown at 37 °C and 5% $CO_2$ until colonies formed (~3 weeks). To screen cells for HiBiT insertion, replica plates were generated by transferring 50 μL of cell suspension to a new plate. Luminescence in the replica plated cells was immediately measured using the lytic HiBiT detection assay (described above), and luminescence positive clones from the corresponding well in the parental plate were expanded.

### ddPCR analysis of HiBiT and HaloTag-HiBiT integration

Genomic DNA was purified by warming the frozen cell pellets to room temperature and adding 50-100 uL of PBS, vortexing, and transferring the cells into cartridges of the Maxwell RSC Cultured Cells DNA Kit (Promega). DNA was quantified with the NanoDrop spectrophotometer (Thermo Fisher Scientific) or QuantiFluor ONE dsDNA System (Promega) using QuantiFluor ONE Lambda DNA (Promega) as a DNA standard, and a GloMax Discover Microplate Reader (Promega GM3000); all DNA samples were diluted to 2-10 ng/μL for ddPCR.

ddPCR was performed using a QX200 Droplet Digital PCR System (Bio-Rad) or a QX ONE Droplet Digital PCR System (Bio-Rad) following the manufacturer's recommendations. PCR conditions employed the ddPCR Supermix for Probes (No dUTP) (BioRad), and the amount of DNA was about 25-50 ng purified genomic DNA per reaction for pools and 2–10 ng DNA for clones. PCR conditions were modified for larger amplicons utilising Subcycling-PCR[83,84] (PCR condition: HiBiT integration (95 °C 30 s, 50x (95 °C 15 s, 4x (62 °C 15 s, 72 °C 30 s) 72 °C 5 min, 4 °C hold)); HaloTag-HiBiT integration (25 °C 1 min, 95 °C 10 min, 50x (95 °C 10 s, 62 °C 15 s, 72 °C 3 min 30 s, 62 °C 15 s, 72 °C 3 min, 62 °C 15 s, 72 °C 3 min) 4 °C 5 min, 25 °C 2 min). Primer and probes are shown in Supplementary Data 2). HiBiT knock-in pool detection assay included two gene-specific primers spanning the targeted site and one gene-specific probe outside the homology arms. Such as, both KI and non-inserted allele produces a fluorescence signal. Additionally, a HiBiT detection probe is added, only producing fluorescence signals in KI alleles. KI levels were calculated by the ratio of HiBiT-to-gene-specific signal. For HaloTag-HiBiT detection, the assay contained two gene-specific primers outside of the homology arms, paired with two primers at the 3′ and 5′ end within the HaloTag encoding region. A fluorescent gene probe was used to detect the gene coding region in the 5′ flank amplicon, and a fluorescent probe to HiBiT was used to detect the 3′ flank amplicon. Thus, a KI allele of HaloTag generates a signal with the gene-specific probe from the amplicon with 5′ gene-specific primer and the 3′ end HaloTag primer, and with the HiBiT probe in the amplicon of the 5′ end HaloTag primer and the 3′ gene-specific primer. Alleles that do not receive a KI will only give a signal with the gene-specific probe and the 5′ gene-specific primer and the 3′ gene-specific primer. All signals were compared to that of $RPPH1$ on chromosome 14, which is expected to be diploid, and therefore the ratio of the 5′ flank or 3′ flank positive droplets would represent the percentage KI, expressed as percent, of HaloTag-HiBiT into the genome. For the detection of HaloTag-HiBiT in cell clones, the ratio of HiBiT to the gene-specific signal represents the number of chromosomes with a KI for each clone and was used to establish whether each clone was heterozygote or homozygote for HaloTag-HiBiT.

### GFP integration in $TRAC$ locus of primary human CD3+ T cells

Integration of GFP expression cassette into the $TRAC$ locus was performed as described previously[85]. CD3+ T cells were activated for 72 h with ImmunoCult T cell activator (STEMCELL Technologies) following the manufacturer's instructions. Cells were treated with inhibitors or DMSO 5 hours before electroporation. RNPs were produced by combining 100 pmol of synthetic sgRNA (Merck; resuspended in TRIS-EDTA (pH 7.5)) with 40 pmol recombinant $Sp$Cas9 protein at room temperature for 30 min using a 2.5:1 molar ratio of sgRNA to $Sp$Cas9. Immediately prior to electroporation, CD3+ T cells were pelleted by centrifugation at 300×$g$ for 5 min, washed once in PBS and

resuspended to $1 \times 10^5$ cells/µL in Lonza electroporation buffer P3. In all, $1 \times 10^6$ T cells were mixed with the CRISPR/Cas9 RNP and dsDNA HDRT (3 µg) in 25 µL total volume in a 96-well v-bottomed plate before being electroporated using the Lonza 4D 96-well electroporation system, pulse code EH-100. After electroporation 125 µL pre-warmed medium was added to each well of the electroporation plate and cells were carefully transferred to culture vessels. Cells were either cultured in the presence of DMSO or inhibitors. In all, 24 h post electroporation medium was changed to remove inhibitors and 7 days later KI frequencies were determined through GFP detection by flow cytometry using the BD LSR-Fortessa Cell Analyzer (BD Bioscience) and FlowJo v10.8.0. At the same timepoint cell viability and live cell counts were determined using the Cellometer Auto 2000 (Nexcelom).

## ddPCR translocation assay

The translocation assay targets *PCSK9* and *HBEGF* and was performed as described previously[86]. In short, genomic DNA was isolated from HEK293T cells 3 days after transfections using the Gentra Puregene Cell Kit (Qiagen) and was diluted to 10 ng/µL. Custom ddPCR assays were ordered from Bio-Rad detecting balanced translocations between *PCSK9* and *HBEGF* (Supplementary Data 2). *AP3B1* (BioRAD) was used as reference assay. ddPCR PCR reaction contained 1× ddPCR Supermix for Probes, no dUPT (Bio-Rad), 1× FAM-labelled *HBEGF-PCSK9* custom assay (BioRAD), 1× *AP3B1*-HEX labelled human reference assay (Bio-Rad), 1/40 *Hae*III (Invitrogen) and 50 ng/µL genomic DNA. 20 µL PCR reaction was used to generate lipid droplets with an automated Droplet Generator (Bio-Rad). PCR amplification was performed using the following conditions: 95 °C for 10 min, 40x (94 °C for 30 s, ramp 2 °C/s; 63.2 °C for 1 min) followed by enzyme deactivation at 98 °C for 10 min. Readout was performed with QX 100 Droplet Reader (Bio-Rad) and ddPCR Droplet Reader Oil (Bio-Rad). Data analysis was conducted with QuantaSoft 1.7.4 Software from Bio-Rad.

## Deep-targeted amplicon next-generation sequencing

Genomic DNA was extracted three days after transfection with QuickExtract (QE) (Lucigen). Cells were washed once with PBS and 50 µL QE was added per well of a 96-well plate. The reaction was incubated for 10 min at 70 °C followed by 10 min at 98 °C. Plates were spun down at maximum speed and stored at −20 °C. Deep-targeted amplicon sequencing was performed from genomic DNA using the NextSeq platform (Illumina). In brief, 1–3 µL genomic DNA from QE, or 50 ng of purified genomic DNA, was used to generate amplicons flanking the CRISPR edited sites with two sequential rounds of PCR. In the first round of PCR forward and reverse sequencing adaptors were introduced with the amplicon-specific primers (Supplementary Data 2). Amplicons were generated with Phusion Flash High-Fidelity PCR Master Mix (Thermo Fisher Scientific) in a 15 µL reaction containing 250 nM of target-specific primers using the following cycling conditions: 98 °C for 3 min, 30-35x (98 °C for 10 s, 60 °C for 5 s, 72 °C for 5 s). For off-target analysis and HiBiT integration Q5 Hot Start High-Fidelity 2x Master Mix (New England Biolabs) was used in 15 µL reactions with 500 nM of target-specific primers and the following cycling conditions for off-target analysis: 98 °C for 3 min, 30x (98 °C for 10 s, 65 °C for 15 s, 72 °C for 15 s), final extension at 72 °C, 2 min and for HiBiT integration 98 °C for 30 s, 25x (98 °C for 10 s, 67 °C for 50 s, 72 °C for 120 s). PCR products were bead-purified using HighPrep PCR Clean-up System (Magbio Genomics) or ProNex Size-Selective Chemistry (Promega) and analysed on a fragment analyser (Agilent) to determine size and concentration. 0.5 ng of PCR1 product was subjected to another round of PCR to add unique Illumina indexes (Nextera XT Index Kit, Illumina) with KAPA Hifi Hotstart Ready Mix (Roche) in a 25 µL reaction including 500 nM indexing primers. Thermocycling conditions were: 72 °C for 3 min, 98 °C for 30 s, 10x (98 °C for 10 s, 63 °C for 30 s, 72 °C for 3 min), 72 °C for 5 min. For HiBiT integration analysis, indexing PCR was performed with Q5 High-Fidelity 2X Master

Mix applying the following cycling conditions: 98 °C for 30 s, 10x (98 °C for 10 s, 65 °C for 75 s, 72 °C for 120 s). Indexing PCR products were bead purified. Purity and average length were analysed with fragment analysis (Agilent), and concentration was quantified with a QuBit 4 Fluorometer (QuBit dsDNA HS Assay Kit, Life Technologies, Thermo Fisher Scientific) or by qPCR (ProNex NGS Library Quant Kit, Promega). DNA libraries were sequenced on Illumina NextSeq 500 or Illumina MiSeq platforms.

## Long-read sequencing for large deletion assessment

HEK293T cells were transfected with different combinations of *Sp*Cas9, *HBEGF*-targeting sgRNA and HDR-plasmid repair template or ssDNA as described above. Three days after transfection the cells were subcultured in a ratio of 1:3. Genomic DNA was isolated one day later with Gentra Puregene Cell Kit (Qiagen). Amplicons for long-range sequencing were generated with Q5 High-Fidelity polymerase in 100 µL reactions (New England Biolabs) including 500 nM primers and up to 1000 ng genomic DNA using the following PCR protocol: 98 °C for 30 s, 30x (98 °C for 10 s, 70 °C for 10 s, 72 °C 6 min) and final extension at 72 °C for 6 min. Target site specific primers contained barcoded adaptors for sample multiplexing (Supplementary Data 2). PCR products were bead-purified using HighPrep PCR Clean-up System (Magbio Genomics) and loaded on a fragment analyser (Agilent) to determine size and concentration. 110 ng of each sample was pooled, and long-read sequencing was performed by GeneWiz/Azenta using PacBio Sequel I SMRT sequencing.

## Bioinformatic analysis

Demultiplexing of Amp-Seq data was performed with bcl2fastq software. The fastq files were analysed with CRISPResso2 version 2.1.1[43] using a quantification window of 8. A detailed list of parameters, such as reference sequences, can be found in Supplementary Data 4. Variant tables from CRISPResso2 analysis were exported into Microsoft Excel files using a Python and R script available in the supplementary information. Variant tables were investigated with RIMA[34] v2, a Microsoft Excel for Microsoft 365 based tool, which uses visual basic programming for applications (VBA) code to analyse and visualise InDel profiles from NGS data. CRISPResso2 and RIMA v2 data analyses comparison is shown in Supplementary Fig. S3a.

Long-range sequencing reads were demultiplexed and subreads were converted into circular consensus sequences (CSS) using PacBio ccs version 6.4.0. Afterwards, reads were aligned to the reference sequence (Supplementary Data 4) using pbmm2 version 1.9.0 and minimap2 version 2.15 with non-default options –preset CCS –sort[87]. Genomic coverage was calculated applying bedtools version 2.20.0 genomecov for raw coverage and deeptools version 3.5.1 bamCoverage for CPM-normalised coverage with non-default options–binSize 1–normalizeUsing CPM[88,89]. Coverage summary statistics were calculated with samtools version 1.15.1 coverage. Visualisation of coverage was performed with JBrowse 2 version 1.7.6[90]. Deletion proportion was calculated as previously described[31], i.e. (Ct−Cm/Ct), where Ct is the number of unique reads that aligned within the amplicon and Cm is the mean per-base read coverage of the amplicon region. Deletion index[31] is the difference in deletion proportion between treated and control groups, as defined previously.

## Statistics and reproducibility

All sequencing experiments were performed with cells separated into three stocks, each individually transfected with the same transfection mix, recovered, and analysed (technical replicate); or cells from separately cultured stocks were individually transfected with different transfection mix, recovered and analysed (biological replicate). Graphical visualisation and statistical analysis were performed with GraphPad Prism 8.4.3 (GraphPad Software, Inc) or JMP 16.1.0 (SAS Institute Inc.). Applied statistical tests, samples sizes and *p*-values are

indicated in the figure legends. No statistical method was used to predetermine sample size. Few data were excluded from the analyses due to low transfection efficiency, as indicate in the reporting summary. The experiments were not randomised. The Investigators were not blinded to allocation during experiments and outcome assessment.

## Reporting summary

Further information on research design is available in the Nature Portfolio Reporting Summary linked to this article.

## Data availability

Next-generation sequencing data are available in the NCBI Sequence Read Archive database (SRA) under BioProject accession code PRJNA907774. Source data are provided with this paper.

## Code availability

The RIMA v2 file is available in the supplementary macro-enabled Excel file. Supplementary Code 1 contains a Python script to process CRIS-PResso2 runs into.txt files for RIMA v2. Supplementary Code 2 comprises an R script to convert.txt files to.xlsx files and adds file paths to the RIMA v2 template. Supplementary Note 2 describes how to use the different scripts. The code is freely available under the MIT license. All files are accessible in the Supplementary Software zipped folder.

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

## Acknowledgements

We thank the AstraZeneca Discovery Sciences Genome Engineering team for fruitful discussions and Mike Snowden for supporting this work. We thank Anna Forslöw for her advice with PHH and the NGS & Transcriptomic team for their support with deep targeted amplicon-sequencing. We acknowledge Kevin Holden and Synthego for their support with KO cell. This project has received funding from the European Union's Horizon 2020 research and innovation programme under the Marie Skłodowska-Curie grant agreement no. 765269 (S.W.). A.L., S.L., S.C. and B.B. were postdoc fellows of the AstraZeneca R&D postdoc program during this study.

## Author contributions

S.W., A.T.G. and M.M. conceptualised the study. S.W. and N.A. performed most of the experimental work with the help from A.L., P.H., S.L., S.C., B.B., B.M. S.DC, P.I, S.Š., P.A. and A.T.G.; The TLR screen was performed by J.B., S.E. and O.E. and supervised by T.N. HiBiT and HaloTag-HiBiT experiments were performed by M.K.S., M.R.S. and T.M.; M.F. bioinformatically analysed NGS Amp-Seq data and B.S. performed analysis of long-range sequencing data. A.T.G. developed RIMA v2. S.W. prepared the manuscript with input from N.A., S.Š., P.A. and M.M. and reviews from other authors. M.B, E.B.C., J.V.F. and S.R. helped with revision and provided scientific guidance. M.M. supervised the study.

## Competing interests

S.W., N.A., M.F., J.B., S.E., A.L., P.H., S.L., S.C., J.S., B.B., B.S., B.M., S.DC., P.I., M.B., T.M., S.R., O.E., E.B.C., J.V.F., S.Š., P.A., A.T.G. and M.M. are presently or were previously employed by AstraZeneca and may be AstraZeneca shareholders. M.K.S., M.R.S. and TM are presently employed by Promega Corporation. S.W., N.A., S.Š. and M.M. are listed as co-inventors in an AstraZeneca patent (WO2023052508A2) related to this work.

## Additional information

[1]Genome Engineering, Discovery Sciences, BioPharmaceuticals R&D, AstraZeneca, Gothenburg, Sweden. [2]Department of Chemistry & Molecular Biology, University of Gothenburg, Gothenburg, Sweden. [3]Data Sciences & Quantitative Biology, Discovery Sciences, BioPharmaceuticals R&D, AstraZeneca, Cambridge, UK. [4]Cell Assay Development, Discovery Sciences, BioPharmaceuticals R&D, AstraZeneca, Gothenburg, Sweden. [5]Cell Engineering Sweden, Discovery Sciences, BioPharmaceuticals R&D, AstraZeneca, Gothenburg, Sweden. [6]Promega Corporation, Madison, WI, USA. [7]Translational Genomics, Discovery Sciences, BioPharmaceuticals R&D, AstraZeneca, Gothenburg, Sweden. [8]Translational Science & Experimental Medicine, Research and Early Development, Respiratory & Immunology (R&I), BioPharmaceuticals R&D, AstraZeneca, Gothenburg, Sweden. [9]Cell Immunology, Discovery Sciences, BioPharmaceuticals R&D, AstraZeneca, Cambridge, UK. [10]Data Sciences & Quantitative Biology, Discovery Sciences, BioPharmaceuticals R&D, AstraZeneca, Gothenburg, Sweden. [11]Compound Synthesis & Management, Discovery Sciences, BioPharmaceuticals R&D, AstraZeneca, Gothenburg, Sweden. [12]Molecular AI, Discovery Sciences, BioPharmaceuticals R&D, AstraZeneca, Gothenburg, Sweden. [13]Discovery Sciences, BioPharmaceuticals R&D, AstraZeneca, Cambridge, UK. [14]Bioscience, Early Oncology, AstraZeneca, Cambridge, UK. ✉e-mail: sandra.wimberger@astrazeneca.com; marcello.maresca@astrazeneca.com

