## [Peer Review File · Nature Communications]

Reviewers' Comments:

Reviewer #1:

Remarks to the Author:

In this manuscript Wimberger et al. use a reporter cell-line and perform a large-scale compound screen to find targets whose inhibition improves the efficiency of CRISPR/Cas9-based precise gene editing. They found the previously identified and characterized DNA-PK inhibitor AZD7648 as main hit, which they then further characterize in its ability to modulate the outcome of CRISPR/Cas9-induced editing, initially by itself, but later in the manuscript the authors also combine DNA-PK inhibition with two novel compounds that inhibit Pol θ to block MMEJ/TMEJ. They conclude that combined inhibition of these two end-joining components strongly increases the frequency of HDR, a strategy they therefore name 2iHDR.

Overall, the experiments are carefully described and proper controls are included. The inclusion of data sets for several human cell lines, including primary cells and iPS cells, nicely validates the utility of the approach. The most pertinent issue with the current manuscript is that it is very similar to recently published work by the Tijsterman lab (Schimmel et al., PMID: 36701230). That said, the data presented here nicely validates the data from Schimmel et al., and takes it one step further by also revealing reduced off-target mutagenesis and reduced translocation frequencies. The authors also report the absence of an effect of Novobiocin, which other labs also found (published and unpublished), which is important given how much research is currently going on in this field of research.

While this "lack of novelty" simply because of unlucky timing may lead to editorial concerns, I would strongly support publication of the paper (after addressing the concerns listed below). The work was obviously performed completely independently, and the conclusions of both studies (i.e. that precise gene-targeting can be boosted) will lead to many in the community following up. Independent confirmation is thus of high value.

I suggest the following points to potentially improve the manuscript

1. To generate optimal impact for the strategy, I would suggest to refer to the recently published work by Schimmel et al., would could have a form as in "while this work was submitted for publication...." Or something of that kind.
2. In their description of the reporter-cell line and the repair outcomes, in the text as well as in Figure 1, the authors refer to mutagenic repair as being 'NHEJ'. I would strongly recommend to change this to 'EJ', as they later show (and many others have shown) that at least MMEJ/TMEJ also plays an important role in the mutagenic repair of CRISPR/CAS9-induced breaks in contexts.
3. The two novel Pol θ -inhibitors (PolQ1 and PolQ2) are very poorly described and characterized. The authors should provide more information on what these inhibitors exactly are and do; additional validation (e.g. checking for additional affects in Polq-knockout cells and in vitro inhibition of the polymerase) would also be in place.
4. Introduction, lines 71 and following. The authors need to be more precise in their wording. I would first of all consider using TMEJ instead of MMEJ for when MMEJ mediated by pol theta is demonstrated/assayed, for three arguments: i) the Ramsden Lab has shown that NHEJ also uses microhomology, ii) the Stark and Tijsterman labs have shown that MMEJ can be independent of Pol theta when the microhomology is a bit more substantial (than ~10bp), and iii) yeast performs MMEJ but does not encode pol theta. As the authors inhibit Pol theta with a specific inhibitor, hence impeding pol theta mediated end-joining, I would favor a more precise term. Secondly, they report a specific role for PARP in MMEJ. I feel that that conclusion cannot be derived from the referenced study, while meanwhile a/the role for PARP in MMEJ is far from clear. I would suggest removing this statement altogether as it is not needed. Finally, it is not known whether Pol theta "performs gap filling". The current data support a role for Pol theta in extending 3' ends on a template that came about by annealing of complementary bases, but it is an outstanding question which polymerase fills the gap, which could easily be pol delta after the MH is extended such that it provides a primer for a replicative error-free polymerase.
5. Introduction, line 83-84: The authors claim that there is 'no other strategy besides HDR that is capable of efficiently and precisely installing different genomic modifications, including'. That is incorrect: strategies like base-editing and prime-editing do not rely on HDR and can install specific genomic modifications. Likewise, a recent paper has described the insertion of large sequences

without the need of HDR (PMID: 36424489)

6. Lines 94-96: consider including Wyatt et al. (PMID: 27453047) and Schimmel et al. (PMID: 29079701) to the references here as these studies also have shown robust involvement of Pol-Theta during the repair Cas9-induced DSBs.

7. Lines 99-100: ART558 has been further tested and validated (PMID: 36701230; PMID: 36200480)

8. In most figure panels 'editing efficiency' is used as label on the y-axis. This is, however, only based on the products that can be picked up by the short-read amplicon sequencing approach (e.g. missing large deletions). Consider changing this to 'mutagenic reads' or 'relative editing efficiency' might be more appropriate.

9. Lines 298-302 and figures 5b/S8: I could not find a definition of 'large deletions', when is a deletion considered to be large?

10. And about this data on large deletions: it might be informative to include a microhomology(MH)-usage plot for these larger deletions. I observe a reduction of large deletions upon PolQ inhibition, but are those that are left under these conditions for example depleted for MH-usage? Or is there any overrepresentation of larger MH-stretches, which could point to extended-MH/SSA kind of mechanism for their formation?

11. How does the KI-Seq pipeline differ (or compare) to other methods used to infer mutational signatures at CRISPR/Cas9 induced breaks (e.g. SiQ, PMID: 36071722; CRISPResso2, PMID: 30809026 and AmpliCan, PMID: 30850374)

Reviewer #2:

Remarks to the Author:

The authors have screened a library of over 20K compounds for those that promote precision gene editing. The top hits are inhibitors of DNA-PK, a protein kinase required for NHEJ. Previous literature has shown that inhibition of DNA-PK inhibits NHEJ by blocking autophosphorylation induced release of DNA-PKcs from DSB ends. Accordingly, DNA-PK inhibitors block NHEJ and promote HDR which promotes precision gene editing. Moreover, the authors find that combination of inhibition of DNA-PK with inhibition of pol theta, which is required for the alternative and potentially mutagenic TMEJ pathway, further improves efficiency of gene editing. The findings have significance for gene editing and it's various applications.

The paper is well written and the authors test a variety of DSB repair pathway inhibitors in a variety of cancer cell lines (Hek293T, Jurkay and HepG2) as well as non-transformed human cells (iPSC, primary T cells and primary human hepatocytes). Results are shown as mean \pm SD from 3 biological replicates.

Minor comments:

Figure 5C: please indicate statistical significance indicated by ****

Line 538 and Sup Figure 1b: Please provide reference for KU60648 (according to the literature it is a dual PI3K/DNA-PK inhibitor) and explain why it was used as a positive control rather than a more specific DNA-PK inhibitor.

Reviewer #3:

Remarks to the Author:

The paper by Wimberger et al focuses on the method of improvement of the efficiency and precision of genome editing. The authors identified DNA-dependent Protein Kinase (DNA-PK) as the most effective target for improving CRISPR/Cas9-mediated genome insertions and demonstrated that a DNA-PK inhibitor AZD7648 enhances accurate CRISPR/Cas9-mediated integration by increasing HDR and decreasing mutagenic NHEJ repair. They also found that additional improvement can be achieved by a combination DNA-PK inhibitor AZD7648 with inhibition of Polymerase Theta.

The quality of experimental work is high; experiments are expertly designed and executed. The

major results are reproduced in several cell lines. The presentation quality is also sufficiently high.

The study is technologically advanced employing a variety of quite sophisticated techniques, like KI-sequencing. However, at the conceptual level the results of this study confirm either previous findings or intuitive expectations. For instance, it was known for quite a long time that inhibition/suppression of NHEJ leads to an increase in HDR. Overall, while the paper may probably be published in Nature Communication, a sister journal Nature Biotechnology would be a better fit.

Response to reviewers:

Wimberger S. *et al.* - Simultaneous inhibition of DNA-PK and Pol Θ improves integration efficiency and precision of genome editing.

We thank all the reviewers for reviewing our manuscript for publication in *Nature Communications* and providing us with their valuable comments. We were delighted to receive positive feedback from the reviewers regarding our work. Herein, we present our point-by-point response to the comments and highlight the new information that we have incorporated to enhance the quality of our manuscript. The revised manuscript includes a detailed account of the additions and corrections made (highlighted in yellow). We appreciate the reviewers' efforts in helping us refine our work.

Reviewer #1 (Remarks to the Author):

In this manuscript Wimberger *et al.* use a reporter cell-line and perform a large-scale compound screen to find targets whose inhibition improves the efficiency of CRISPR/Cas9-based precise gene editing. They found the previously identified and characterized DNA-PK inhibitor AZD7648 as main hit, which they then further characterize in its ability to modulate the outcome of CRISPR/Cas9-induced editing, initially by itself, but later in the manuscript the authors also combine DNA-PK inhibition with two novel compounds that inhibit Pol Θ to block MMEJ/TMEJ. They conclude that combined inhibition of these two end-joining components strongly increases the frequency of HDR, a strategy they therefore name 2iHDR.

Overall, the experiments are carefully described and proper controls are included. The inclusion of data sets for several human cell lines, including primary cells and iPS cells, nicely validates the utility of the approach. The most pertinent issue with the current manuscript is that it is very similar to recently published work by the Tijsterman lab (Schimmel *et al.*, PMID: 36701230). That said, the data presented here nicely validates the data from Schimmel *et al.*, and takes it one step further by also revealing reduced off-target mutagenesis and reduced translocation frequencies. The authors also report the absence of an effect of Novobiocin, which other labs also found (published and unpublished), which is important given how much research is currently going on in this field of research.

While this "lack of novelty" simply because of unlucky timing may lead to editorial concerns, I would strongly support publication of the paper (after addressing the concerns listed below). The work was obviously performed completely independently, and the conclusions of both studies (i.e. that precise gene-targeting can be boosted) will lead to many in the community following up. Independent confirmation is thus of high value.

We thank Reviewer #1 for thoughtfully evaluating our manuscript and supporting its publication. We also appreciate the reviewer's comment regarding the work by Schimmel *et al.* (PMID 36701230), which was published while we submitted our study for publication. Their study shows that inhibiting Pol Θ with ART558 improves the safety of genome editing and promotes precise genome editing through HDR. We were excited to see an independent confirmation of our findings on increasing integration efficiency by temporarily inhibiting two key enzymes of EJ repair. While Schimmel *et al.* focus on a more refined analysis of the formation of large deletions, we further expand the safety aspect by showing that 2iHDR reduces off-target editing and translocations, as pointed out by the reviewer. Furthermore, our work provides a comprehensive evaluation of small molecules, resulting in the following:

1. the identification of a more potent and selective DNA-PK inhibitor (AZD7648);
2. the validation of two additional potent Pol Θ inhibitors besides ART558; and

3. the subsequent observation that inhibiting either the polymerase or helicase domain of Pol Θ enhances precise genome editing.
Thus, the study by Schimmel et al. and our work significantly strengthen and complement each other.

Likewise, we appreciate the diligent and critical review of this study by reviewer #1. By addressing their concerns and suggestions, we believe the current manuscript is a more precise and improved version.

I suggest the following points to potentially improve the manuscript

1. To generate optimal impact for the strategy, I would suggest to refer to the recently published work by Schimmel et al., would could have a form as in “while this work was submitted for publication...” Or something of that kind.

We have added the following text to the introduction (Lines 102 - 106):

“While we submitted this work for publication, a study was published demonstrating that the Pol Θ inhibitor ART558 (PMID: 34140467 PMID: 36200480) restrains the formation of large deletions and promotes HDR at CRISPR/Cas9-induced DSBs (PMID: 36701230). However, further investigations of other potential genomic alterations are still lacking. Identifying novel inhibitors with higher specificity and potency might further improve precise genome editing.”

2. In their description of the reporter-cell line and the repair outcomes, in the text as well as in Figure 1, the authors refer to mutagenic repair as being ‘NHEJ’. I would strongly recommend to change this to ‘EJ’, as they later show (and many others have shown) that at least MMEJ/TMEJ also plays an important role in the mutagenic repair of CRISPR/CAS9-induced breaks in contexts.

We appreciate the comment of Reviewer #1. We have modified the text following Reviewer’s comment to terminology encompassing NHEJ and alt-EJ.

We have modified the text and have updated Figure 1 and Supplementary Figure S1.

3. The two novel Pol Θ -inhibitors (PolQi1 and PolQi2) are very poorly described and characterized. The authors should provide more information on what these inhibitors exactly are and do; additional validation (e.g. checking for additional affects in Polq-knockout cells and in vitro inhibition of the polymerase) would also be in place.

We agree with the reviewer's comment that the description of PolQi1 and PolQi2 is limited. Therefore, we have added additional information to our manuscript, which is summarized in Figure R1. The structure of PolQi1 and PolQi2, along with biochemical data reported in their published patents, are displayed in Figure R1a. Moreover, we conducted a study on the efficacy of these compounds on cellular microhomology-mediated deletions without adding a DNA repair template, as shown in Figure R1b and R1c. Finally, we compared the dose-response curves and IC50 values of PolQi1 and PolQi2 to ART558, as presented in Figure R1d and R1e. Our findings indicate that PolQi2 is more potent than PolQi1, which exhibited similar potency to ART558. This outcome is consistent with our data on the dose-dependent increase of KI efficiencies. We have updated the manuscript with the new data by dividing Supplementary Figure S7 into two new figures (Supplementary Figures S7 & S8). We replaced the data in the original Supplementary Figure S7e with the new titration shown in Figure R1e. We have also updated the numbering of the figures.

Figure R1. Description and cellular characterization of PolQi1 and PolQi2. **(a)** Structure of PolQi1 (WO2021/028643, Example 158) and PolQi2 (WO2020/243459, Example 99) and reported biochemical data. **(b&c)** Mutation frequency of different repair events at indicated target sites in HEK293T cells, treated with increasing concentrations of PolQi1 or PolQi2 (0.1 – 10 μ M) and DMSO control. Cells were treated with compounds and transfected with plasmids 1-3 hours later. Bar graphs show mean values \pm SD (n=3, biological replicates). **(d,e)** Dose-dependent effect on (d) MH-deletions and (e) distribution of DNA repair events in gMej-targeted HEK293T cells treated with several Pol θ inhibitors (0.1 – 10 μ M) in the presence of 1 μ M AZD7648. (d) Dose-response curves show mean values \pm SD (n=3, technical replicates). IC₅₀ values were calculated using a 3-parameter model fit. R² and 95% confidence interval (CI) are indicated. (e) Bar graphs illustrate mean percentage of mutated reads \pm SD (n=3, technical replicates).

We modified the results section's text (Lines 281 - 290).

"Supplementary Figure S8b depicts the structure of PolQi1 and PolQi2, along with the biochemical data reported in their published patents. To validate the inhibitors and investigate their efficacy on cellular TMEJ repair, we pre-treated HEK293T cells with different concentrations of PolQi1 or PolQi2. We assessed the mutational profile at three different target sites and found that cells treated with PolQi1 or PolQi2 showed a partial reduction of microhomology-associated deletion, similar to our POLQ knock-out pool data (Figure 4a, Supplementary Figure S7a,b & S8c,d). Lastly, we compared the dose-response curves and IC50 values of PolQi1 and PolQi2 to ART558 in the presence of AZD7648. The IC50 values were calculated based on the response of MH-mediated deletion. We observed similar potency between PolQi1 and ART558, while PolQi2 displayed approximately ten times higher potency (Supplementary Figure S8d)."

4. Introduction, lines 71 and following. The authors need to be more precise in their wording. I would first of all consider using TMEJ instead of MMEJ for when MMEJ mediated by pol theta is demonstrated/assayed, for three arguments: i) the Ramsden Lab has shown that NHEJ also uses microhomology, ii) the Stark and Tijsterman labs have shown that MMEJ can be independent of Pol theta when the microhomology is a bit more substantial (than ~10bp), and iii) yeast performs MMEJ but does not encode pol theta. As the authors inhibit Pol theta with a specific inhibitor, hence impeding pol theta mediated end-joining, I would favor a more precise term. Secondly, they report a specific role for PARP in MMEJ. I feel that that conclusion cannot be derived from the referenced study, while meanwhile a/the role for PARP in MMEJ is far from clear. I would suggest removing this statement altogether as it is not needed. Finally, it is not known whether Pol theta "performs gap filling". The current data support a role for Pol theta in extending 3' ends on a template that came about by annealing of complementary bases, but it is an outstanding question which polymerase fills the gap, which could easily be pol delta after the MH is extended such that it provides a primer for a replicative error-free polymerase.

We appreciate the comment of the reviewer and agree with this point. Therefore, we have removed the statement about Parp1, used more precise wording by introducing alt-EJ and TMEJ, and revised the section as follows (Lines 71 - 81):

"In contrast, alt-EJ is inherently prone to errors and has the potential to create larger deletions. Alt-EJ typically uses short microhomology sequences flanking the break site and requires 3'-single-stranded DNA substrates, which are generated by nucleolytic processing of DNA, a process termed end resection. Though the initial steps of end resection are shared with HDR, limited end resection is sufficient to promote alt-EJ. DNA Polymerase Theta (Pol Θ) is the primary but not an exclusive mediator of alt-EJ in most eukaryotic cells. Therefore, alt-EJ through Pol Θ is referred to as Pol Θ -mediated end-joining (TMEJ). The helicase domain of Pol Θ most likely promotes the annealing of resected 3' overhangs utilizing microhomologies, while the polymerase domain extends annealed sequences. Resolution involves flap removal through endonucleases, such as Flap Endonuclease 1 (FEN1) 17, gap filling, and, finally, joining of ends by DNA ligase 1 (LIG1) or DNA ligase 3 (LIG3)."

We added a statement to the result section to explain how we assess alt-EJ/TMEJ repair and the limitation of this definition (Lines 170 - 173).

"We utilized microhomology-mediated deletions as a substitute to evaluate alt-EJ/TMEJ, disregarding other alt-EJ events such as synthesis-dependent alt-EJ (PMID:

29121353). Additionally, it should be noted that microhomology-mediated deletion can also result from NHEJ repair (PMID: 34522048).”

We have also updated the abstract, introduction, results, and discussion sections, along with the figures and supplementary figures.

5. Introduction, line 83-84: The authors claim that there is ‘no other strategy besides HDR that is capable of efficiently and precisely installing different genomic modifications, including ..’. That is incorrect: strategies like base-editing and prime-editing do not rely on HDR and can install specific genomic modifications. Likewise, a recent paper has described the insertion of large sequences without the need of HDR (PMID: 36424489)

We thank Reviewer #1 for bringing this to our attention. Although, in our initial draft, we mention that “...no other strategy besides HDR is currently capable of installing different genomic modifications, including ...” we agree that this statement is misleading. Therefore, we remove the claim that “no other strategy is capable of” and have modified the sentence as follows (Lines 86 - 89).

“However, the flexibility of HDR, in enabling scarless installation of various genomic modifications, such as point mutations, deletions, and kilobase insertions, makes it an essential modality for research and clinical developments. Therefore, developing strategies for bias repair towards HDR is crucial for precise genome editing applications.”

6. Lines 94-96: consider including Wyatt et al. (PMID: 27453047) and Schimmel et al. (PMID: 29079701) to the references here as these studies also have shown robust involvement of Pol-Theta during the repair Cas9-induced DSBs.

We thank the reviewer for pointing out critical references that we have implemented in the text (Line 99).

7. Lines 99-100: ART558 has been further tested and validated (PMID: 36701230; PMID: 36200480)

We agree with the reviewer that ART558 has been tested and validated. However, the statement “...require further testing and validation...” along with the cited reference for ART558, is incorrect. Therefore, we have removed the statement and updated the following text (Line 101 - 102).

“However, small molecule inhibitors of Pol θ have only recently started to emerge (PMID: 34140467, PMID: 36126059, PMID: 34179826).”

8. In most figure panels ‘editing efficiency’ is used as label on the y-axis. This is, however, only based on the products that can be picked up by the short-read amplicon sequencing approach (e.g. missing large deletions). Consider changing this to ‘mutagenic reads’ or ‘relative editing efficiency’ might be more appropriate.

We agree with the reviewer that the current axis title is misleading. We have modified the y-axis title of Figure 2, 3, 4, 5 & Supplementary Figure S3, S7, S8 to “Mutated reads (%)” and “Reads with knock-in (%)”.

9. Lines 298-302 and figures 5b/S8: I could not find a definition of ‘large deletions’, when is a deletion considered to be large?

We want to thank the reviewer for their question and provide further explanation. While an earlier study defined large deletions as extending more than 100 bp (PMID:

29562890), we did not provide a specific range for a large deletion. Instead, we assessed the effect of dual inhibition on large deletions using the previously published deletion index (PMID: 34416913), as described in our methods section. This publication reported that the deletion index correlates well with calculating the fraction of deletions over 100 bp. To demonstrate that this is also true for our dataset, we calculated the percentage of large deletions over 100 bp (Figure R2a). Again, the data correlated with the deletion index (Figure R2b).

Figure R2. Assessment of EJ inhibition on large deletions with Long-Read Sequencing: Comparison of deletion index and deletion fraction >100 bp. **(a)** Quantification of large deletions from long-read sequencing data of *HBEGF*-edited HEK293T cells treated with indicated compounds in the following concentrations 1 μ M AZD7648, 3 μ M PolQi1, 3 μ M PolQi2. The bar graphs present the percentage of the fraction of deletions over 100 bp (n=1). We calculated the percentage of reads with deletions more prominent than 100 bp in the amplicon compared to all reads aligned within the amplicon region. We considered the percentage of reads that contain at least one large deletion event relative to all reads within the amplicon coordinates. It was necessary for these reads to overlap with the boundaries of the amplicon, as defined above, by at least 1 bp to be considered. Finally, we subtracted the control group from the edited samples. **(b)** Comparison of deletion index (%) and deletions >100 bp (%) of data from Figure R2a and Figure 5b. The graph shows the Pearson correlation coefficient. The deletion index is the difference in deletion proportion between edited and control groups. The formula $(C_t - C_m / C_t)$ calculates the deletion proportion, where C_t is the number of unique reads that align within the amplicon, and C_m is the mean per-base read coverage of the amplicon region.

10. And about this data on large deletions: it might be informative to include a microhomology(MH)-usage plot for these larger deletions. I observe a reduction of large deletions upon PolQ inhibition, but are those that are left under these conditions for example depleted for MH-usage? Or is there any overrepresentation of larger MH-stretches, which could point to extended-MH/SSA kind of mechanism for their formation?

We thank the reviewer for the suggestion. We believe that the reviewer's question has already been addressed by the detailed analysis performed by Schimmel et al. (PMID 36701230). In this work, the authors did not find a significant overrepresentation of larger MH-stretches in cells that were impaired for both NHEJ and TMEJ repair. While we recognize the importance of examining repair mechanisms linked to large deletions caused by Cas9-induced DSBs, a more comprehensive approach is required to

uncover the underlying mechanisms in those cells, which goes beyond the remit of our manuscript.

11. How does the KI-Seq pipeline differ (or compare) to other methods used to infer mutational signatures at CRISPR/Cas9 induced breaks (e.g. SiQ, PMID: 36071722; CRISPResso2, PMID: 30809026 and AmpliCan, PMID: 30850374)

We thank the reviewer for this question and provided further explanation.

KI-Seq builds upon our previously reported analysis pipeline with RIMA as quantification method (<https://academic.oup.com/nar/article/46/16/8417/5055824>) (PMID: 30032200). In brief, high-throughput sequencing data are aligned to the reference sequence, and variants near the Cas9 targeted site are categorised to reflect editing outcomes. In KI-Seq, we refined how insertions are depicted and reported. However, the source code for analysis of variant tables, based on Excel Visual Basic for Applications (VBA), remains similar to the original version. Furthermore, as mentioned in the results section (Line 165 - 170), our data has now been analysed using CRISPResso2 (PMID: 30809026) and RIMA v2. Since RIMA requires as input variant tables and thus pre-processing of NGS raw data, such as reads quality control, merging paired-reads, mapping to the reference, re-alignment, variant calling and converting variants into tabular formats, these steps can be performed as described in the original paper, or with our additionally provided code that converts the AlleleFrequency.zip files obtained from a CRISPResso2 pooled run into variant Excel tables for analysis RIMA v2.

In the discussion, we pointed out, that unlike most open-access analysis tools, which combine all indels into a single category (referring to CRISPResso2 (PMID: 30809026), Cas-Analyzer (PMID: 27559154) and ampliCan (PMID: 30850374)), KI-Seq provides a more detailed analysis. RIMA categorises, quantifies, and visualises insertions and deletions based on their size and type (e.g., microhomology-associated deletions, duplication), thus providing a more granular analysis. Therefore, RIMA v2 is valuable for a refined analysis of mutations contributing to NHEJ, alt-EJ, and knock-ins from data analysed with CRISPResso2.

Reviewer #2 (Remarks to the Author):

The authors have screened a library of over 20K compounds for those that promote precision gene editing. The top hits are inhibitors of DNA-PK, a protein kinase required for NHEJ. Previous literature has shown that inhibition of DNA-PK inhibits NHEJ by blocking autophosphorylation induced release of DNA-PKcs from DSB ends. Accordingly, DNA-PK inhibitors block NHEJ and promote HDR which promotes precision gene editing. Moreover, the authors find that combination of inhibition of DNA-PK with inhibition of pol theta, which is required for the alternative and potentially mutagenic TMEJ pathway, further improves efficiency of gene editing. The findings have significance for gene editing and it's various applications.

The paper is well written and the authors test a variety of DSB repair pathway inhibitors in a variety of cancer cell lines (Hek293T, Jurkay and HepG2) as well as non-transformed human cells (iPSC, primary T cells and primary human hepatocytes). Results are shown as mean \pm SD from 3 biological replicates.

We thank the reviewer for evaluating our manuscript. We are pleased to hear that the manuscript is well written. The acknowledgement of 2iHDR being of interest to the genome editing community is greatly appreciated.

Minor comments:

Figure 5C: please indicate statistical significance indicated by ****

We have now indicated the statistical significance in Figure 5c.

Line 538 and Sup Figure 1b: Please provide reference for KU60648 (according to the literature it is a dual PI3K/DNA-PK inhibitor) and explain why it was used as a positive control rather than a more specific DNA-PK inhibitor.

We have now added a reference (PMID 22576130) for KU0060648 (Line 559).

We thank the reviewer for their question. While developing the TLR assay and screening for a reference compound to increase HDR and decrease EJ, we did not specifically search for a DNA-PK inhibitor as a control. Instead, we selected compounds based on their robustness, assay window, and low toxicity. We initially used Nu7026 (PMID 14522929) as a tool compound but later compared it to Nu7441 (PMID: 15546735) and KU600648 (PMID 22576130). Fig. R3a shows the dose-dependent efficacy of these compounds in the TLR assay. We chose KU0060648 due to its better assay window at the required concentration and maximum effects on HDR and EJ, as normalized to NU7026 (Fig. R3ab).

Figure R3. KU600648 proved to be an adequate reference compound for the TLR screen. a) Concentration gradient analysis of DNA-PK inhibitors with varying potencies and selectivities to determine their effect on HDR and EJ in the TLR assay. b) Table comparing effects of inhibitors measured in the TLR assay. The bracket shows a repeated measurement.

Reviewer #3 (Remarks to the Author):

The paper by Wimberger et al focuses on the method of improvement of the efficiency and precision of genome editing. The authors identified DNA-dependent Protein Kinase (DNA-PK)

as the most effective target for improving CRISPR/Cas9-mediated genome insertions and demonstrated that a DNA-PK inhibitor AZD7648 enhances accurate CRISPR/Cas9-mediated integration by increasing HDR and decreasing mutagenic NHEJ repair. They also found that additional improvement can be achieved by a combination DNA-PK inhibitor AZD7648 with inhibition of Polymerase Theta.

The quality of experimental work is high; experiments are expertly designed and executed. The major results are reproduced in several cell lines. The presentation quality is also sufficiently high.

The study is technologically advanced employing a variety of quite sophisticated techniques, like KI-sequencing. However, at the conceptual level the results of this study confirm either previous findings or intuitive expectations. For instance, it was known for quite a long time that inhibition/suppression of NHEJ leads to an increase in HDR. Overall, while the paper may probably be published in Nature Communication, a sister journal Nature Biotechnology would be a better fit.

We are delighted to see the enthusiasm of Reviewer #3 for our manuscript and appreciate their positive feedback on the quality of our experimental work. Although our findings may seem intuitive initially, our comprehensive analysis of an extensive compound library helps to fill a gap in the literature, especially with our confirmation of DNA-PK as the best target to increase precise gene editing. Furthermore, we identified a more potent and selective DNA-PK inhibitor, which will further improve gene targeting efficiency and mitigate unwanted cytotoxic effects. Additionally, the genome editing community will benefit from a more profound analysis of recently developed Pol Θ inhibitors, *e.g.* additional cell lines, templates and integration-to-cut-site distances. Finally, our observation of reducing unwanted on- and off-target effects while combining them with NHEJ inhibition will further promote HDR-based genome editing strategies for therapeutic and research purposes.

Reviewers' Comments:

Reviewer #1:

Remarks to the Author:

The authors did an excellent job in revising the manuscript. I have no further suggestions and fully support the publication of this study. Congratulations with this great work!

Reviewer #2:

Remarks to the Author:

The authors have addressed the reviewers concerns

Response to reviewers:

Wimberger S. *et al.* - Simultaneous inhibition of DNA-PK and Pol Θ improves integration efficiency and precision of genome editing.

We thank all the reviewers for reviewing our manuscript for publication in *Nature Communications* and providing us with their valuable comments. We were delighted to receive positive feedback from the reviewers regarding our work. Herein, we present our point-by-point response to the comments.

Reviewers' comments**Reviewer #1 (Remarks to the Author):**

The authors did an excellent job in revising the manuscript. I have no further suggestions and fully support the publication of this study. Congratulations with this great work!

We are delighted to see that Reviewer #1 is enthusiastic about our manuscript and we are grateful for their positive feedback on the revised version. We thank the reviewer for their critical revision and support for publishing our study.

Reviewer #2 (Remarks to the Author):

The authors have addressed the reviewer's concerns.

We are pleased to receive positive feedback from Reviewer #2 regarding our revised manuscript and thank the Reviewer #2 for their effort in evaluating our work.